# Mind the Gap: Structure-Aware Consistency in Preference Learning

**Mehryar Mohri** [1 2]   **Yutao Zhong** [1]

## Abstract

Aligning Large Language Models (LLMs) with human intent, whether through explicit reward modeling or direct methods such as DPO, fundamentally relies on minimizing a surrogate loss as a proxy for the true pairwise ranking objective. We prove that this reliance is flawed for the standard surrogate losses used: for the equicontinuous hypothesis sets characteristic of neural networks, *no* standard surrogate provides a meaningful consistency guarantee. Minimizing the surrogate loss to zero can leave the true ranking error arbitrarily high. To resolve this, we formulate LLM alignment within a margin-shifted ranking framework and derive $\mathcal{H}$-consistency bounds showing that enforcing a confidence margin $\gamma$ is not merely beneficial but *necessary* for consistency. We further introduce Structure-Aware $\mathcal{H}$-consistency and a corresponding objective (SA-DPO) that adapts the margin to the semantic distance between responses, preventing instability on near-synonymous pairs. Finally, we analyze the trade-off between the margin required for consistency and the model's finite capacity to satisfy it, revealing a strict hierarchy of surrogate losses: heavy-tailed surrogates (e.g., the Polynomial Hinge family) offer strictly superior consistency guarantees for capacity-bounded models compared to the logistic loss used in DPO. Experiments on UltraFeedback and Argilla DPO-Mix-7k confirm that SA-DPO consistently outperforms DPO and SimPO, with a 58.5% head-to-head winrate in downstream generation quality.

[1]Google Research, New York, NY; [2]Courant Institute of Mathematical Sciences, New York, NY. Correspondence to: Mehryar Mohri <mohri@google.com>, Yutao Zhong <yutaozhong@google.com>.

*Proceedings of the $43^{rd}$ International Conference on Machine Learning*, Seoul, South Korea. PMLR 306, 2026. Copyright 2026 by the author(s).

## 1. Introduction

The alignment of Large Language Models (LLMs) with human intent relies, at its core, on minimizing a convex surrogate loss over pairwise preference data. This is true both in *explicit* reward modeling, where a reward function is trained on preference pairs before being optimized via reinforcement learning (e.g., PPO; Schulman et al., 2017; Stiennon et al., 2020), and in *implicit* approaches such as Direct Preference Optimization (DPO) (Rafailov et al., 2023), which bypass the reward modeling step entirely. In every case, the surrogate (e.g., the logistic loss) serves as a proxy for the true objective: the non-convex, discontinuous 0-1 ranking loss. This ubiquitous reliance raises a fundamental theoretical question that remains largely unanswered for deep networks: *Does minimizing these surrogate losses actually guarantee the minimization of the true ranking error?*

In this work, we investigate this question through the lens of $\mathcal{H}$-*consistency* (Mao, Mohri, and Zhong, 2023e; 2024h). We formulate LLM preference learning as a pairwise ranking problem and derive a series of results that bridge the gap between learning theory and practical fine-tuning. First, we identify a fundamental theoretical deficiency in standard approaches. We demonstrate that for *equicontinuous* hypothesis sets, a property satisfied by neural networks, standard surrogate minimization yields *vacuous* consistency guarantees. Specifically, without explicit constraints, a model can achieve arbitrarily low surrogate risk while maintaining a high ranking error, effectively "cheating" the objective by shrinking score differences rather than learning the correct ordering.

To resolve this, we introduce the framework of *Margin-Shifted Surrogates*. We prove that enforcing a confidence gap $\gamma$ is not merely a heuristic, but a strict requirement for $\mathcal{H}$-consistency in the deep learning regime. However, while a uniform margin restores consistency, it is a blunt instrument. We show that demanding a large, fixed margin on semantically identical pairs (synonyms) forces the model to hallucinate differences where none exist, introducing bias and instability. To address this, we propose *Structure-Aware $\mathcal{H}$-consistency* and a corresponding objective, *Structure-Aware DPO (SA-DPO)*. SA-DPO dynamically adapts the margin based on the semantic distance between responses (see Figure 1), ensuring the model is strictly penalized for

misranking distinct options while correctly relaxing the constraint for ambiguous or interchangeable pairs.

Finally, we analyze the cost of enforcing these margins on models with finite capacity. We introduce the *Margin-Capacity Profile*, a metric that quantifies the trade-off between theoretical consistency and a model's ability to satisfy constraints. Our analysis reveals a strict hierarchy of loss functions: while the Logistic loss used in DPO has a linear decay profile, losses with heavier tails (such as Squared loss in IPO (Azar et al., 2024) or Cubic Hinge loss) offer superior consistency guarantees for capacity-bounded models.

### Contributions.

1. We prove that unconstrained surrogate minimization on equicontinuous hypothesis sets yields *vacuous* consistency bounds: minimizing the DPO loss provides no guarantee that the true ranking error decreases (Theorem 3.1).

2. We derive $\mathcal{H}$-consistency bounds for margin-shifted surrogates, proving that a confidence gap $\gamma$ is *necessary* for consistency (Theorem 5.2).

3. We introduce SA-DPO, a structure-aware objective that adapts margins to the semantic distance between responses, preventing instability on synonyms while enforcing strict separation on distinct pairs (Theorem 5.6).

4. We analyze the Margin-Capacity Profile, establishing a strict hierarchy where heavy-tailed losses (Cubic Hinge) strictly outperform light-tailed ones (Logistic) for capacity-bounded models.

Our framework also provides a rigorous foundation for recent empirical advances, most notably SimPO (Meng et al., 2024) and SLiC (Zhao et al., 2023). While these methods empirically demonstrated the benefit of hard margins, we prove that such margins are theoretically necessary for consistency. Furthermore, our analysis exposes the limitation of their uniform design: enforcing constant penalties regardless of semantic similarity is suboptimal. By deriving an adaptive margin, SA-DPO is the principled, structure-aware evolution of margin-based alignment.

**Related Work.** Our analysis bridges three distinct lines of inquiry. *Direct Alignment:* While DPO (Rafailov et al., 2023) and IPO (Azar et al., 2024) implicitly optimize preferences, recent empirical works like SimPO (Meng et al., 2024) and SLiC (Zhao et al., 2023) have introduced margin constraints. Our work provides the missing theoretical justification for these margins, proving them necessary for consistency rather than just empirically beneficial. *Ranking Consistency:* Classical consistency results (Bartlett et al., 2006; Duchi et al., 2010) assume universal function approximation. We build on the $\mathcal{H}$-consistency framework (Awasthi

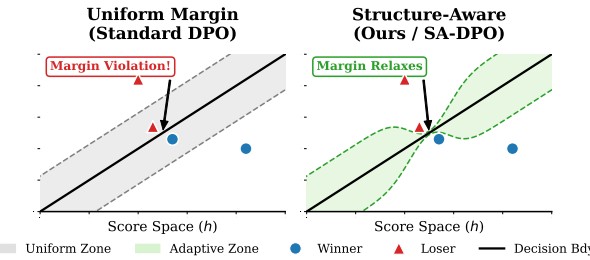

*Figure 1.* Uniform vs. Structure-Aware Margins. (Left) A fixed margin $\gamma$ forces artificial separation on semantically similar pairs (synonyms), causing hallucination. (Right) SA-DPO scales the margin by semantic distance $\Delta(y, y')$, relaxing constraints for synonyms while enforcing strict separation on distinct pairs.

et al., 2022a; Mao et al., 2023c;e) to address the restricted hypothesis sets of deep networks. *Structured Prediction:* Our proposed adaptive margin draws on StructSVMs (Tsochantaridis et al., 2005), extending these ideas to the generative LLM setting. (See Appendix A for an extended discussion).

## 2. Preliminaries

**Setup.** We consider the standard setting of LLM preference learning. Let $\mathcal{X}$ be the space of prompts and $\mathcal{Y}$ be the space of responses. We assume access to a distribution $\mathcal{D}$ over tuples $(x, y, y', w)$ with $y \neq y'$, where $w \in \{-1, 1\}$ indicates the preference ($w = 1$ if $y \succ y'$). Let $\eta(x, y, y') = \mathbb{P}(w = 1 \mid x, y, y')$ be the conditional preference probability. The goal is to learn a scoring function $h \in \mathcal{H}$ (e.g., a reward model or implicit DPO log-ratio) minimizing the *target 0-1 ranking loss* $\mathsf{L}_{0-1}(h, x, y, y', w) = 1_{w \neq \text{sign}(\Delta h)}$, where $\Delta h = h(x, y) - h(x, y')$. We denote the generalization error by $\mathcal{R}(h) = \mathbb{E}[\mathsf{L}_{0-1}]$ and the best-in-class error by $\mathcal{R}^*(\mathcal{H})$. The *conditional error* is $\mathcal{C}(h) = \mathbb{E}[\mathsf{L}_{0-1} \mid x, y, y']$, which induces the *best-in-class conditional error* $\mathcal{C}^*(\mathcal{H}) = \inf_{h \in \mathcal{H}} \mathcal{C}(h)$ and the *conditional regret* $\Delta\mathcal{C}_{\mathcal{H}}(h) = \mathcal{C}(h) - \mathcal{C}^*(\mathcal{H})$. Since optimizing $\mathsf{L}_{0-1}$ is intractable, practical algorithms minimize a convex *surrogate loss* $\mathsf{L}_\Phi(h, x, y, y', w) = \Phi(w \cdot \Delta h)$. For instance, DPO uses $\Phi_{\log}(u) = \log(1 + e^{-\beta u})$ (Chen et al., 2024; Agarwal et al., 2025). We denote the surrogate generalization error by $\mathcal{R}_\Phi(h)$, with conditional terms $\mathcal{C}_\Phi(h)$, $\mathcal{C}_\Phi^*(\mathcal{H})$ and $\Delta\mathcal{C}_{\Phi,\mathcal{H}}(h)$ defined analogously. We define the *minimizability gap* by $\mathcal{M}_\Phi(\mathcal{H}) = \mathcal{R}_\Phi^*(\mathcal{H}) - \mathbb{E}[\mathcal{C}_\Phi^*(\mathcal{H})]$, which measures the difference between the best-in-class error and the expected best-in-class conditional error. It quantifies the approximation error introduced by restricting our model to a specific hypothesis set $\mathcal{H}$ (e.g., neural networks) rather than the space of all measurable functions $\mathcal{H}_{\text{all}}$. As established in recent $\mathcal{H}$-consistency literature (Mao et al., 2024h), when $\mathcal{H} = \mathcal{H}_{\text{all}}$, the minimizability gaps vanish, and our bounds exactly recover the traditional excess risk bounds. A smaller minimizability gap indicates that the restricted hypothesis class $\mathcal{H}$ is well-aligned with the surrogate loss.

**Consistency.** While classical *Bayes-consistency* (Zhang, 2004; Bartlett et al., 2006; Steinwart, 2007) ensures that optimizing a surrogate over all measurable functions minimizes the target loss asymptotically, it provides no guarantees for restricted hypothesis sets $\mathcal{H}$ (e.g., neural networks). We therefore focus on $\mathcal{H}$-*consistency bounds* (Awasthi et al., 2022a; Mao et al., 2023c;e), which provide non-asymptotic guarantees specific to $\mathcal{H}$. A surrogate $\Phi$ is $\mathcal{H}$-consistent if there exists a non-decreasing concave function $\Gamma$ (with $\Gamma(0) = 0$) such that for all $h \in \mathcal{H}$: $\mathcal{R}(h) - \mathcal{R}^*(\mathcal{H}) + \mathcal{M}(\mathcal{H}) \leq \Gamma(\mathcal{R}_\Phi(h) - \mathcal{R}_\Phi^*(\mathcal{H}) + \mathcal{M}_\Phi(\mathcal{H}))$. This bound guarantees that minimizing the surrogate estimation error effectively minimizes the target estimation error, accounting for the approximation limitations captured by the minimizability gaps (Mao et al., 2024h).

## 3. Inconsistency of Unconstrained Ranking

While convenient, surrogate risk minimization does not inherently guarantee consistency with respect to the true target risk. We begin by formally defining the necessary properties for the hypothesis set and then present a negative result of $\mathcal{H}$-consistency.

### 3.1. Equicontinuity and Regularity

We characterize the hypothesis set $\mathcal{H}$ of scoring functions $h : \mathcal{X} \times \mathcal{Y} \to \mathbb{R}$ using two standard assumptions:

1. *Equicontinuity:* For any $\epsilon > 0$, there exist $x, y, y'$ such that $\sup_{h \in \mathcal{H}} |h(x,y) - h(x,y')| < \epsilon$. Equicontinuity implies that the hypothesis class can produce arbitrarily small score differences for certain input pairs (e.g., when responses are near-synonyms). Because modern LLMs represent continuous mappings in a high-dimensional space, they naturally satisfy this property. Note that our definition is a weaker, non-standard condition compared to uniform equicontinuity. We chose this formulation precisely because it makes our negative result (Theorem 3.1) mathematically stronger: consistency completely fails even if the model only produces small score differences on a tiny subset of inputs.

2. *Regularity:* For any tuple $(x, y, y', w)$, there exists $h \in \mathcal{H}$ such that $\text{sign}(\Delta h) = w$. This merely assumes the model class is expressive enough to correctly rank a given pair (i.e., it can achieve the correct sign), which overparameterized LLMs easily satisfy.

These assumptions highlight that without a margin constraint, models can "cheat" by shrinking scores on ambiguous pairs rather than learning robust rankings.

### 3.2. A Vacuous $\mathcal{H}$-Consistency Bound

We now demonstrate that for regular and equicontinuous hypothesis sets, any $\mathcal{H}$-consistency bound is inherently vacuous. Our result and proof technique build upon the $\mathcal{H}$-consistency framework and the fundamental inconsistency findings for pairwise ranking introduced by Mao et al. (2023c, Theorem 2.1). We adapt and extend their analysis to specifically address the equicontinuous hypothesis sets typical of deep networks.

**Theorem 3.1** (**Negative Results for Equicontinuous $\mathcal{H}$).** *Assume that $\mathcal{H}$ is* regular*, equicontinuous, and contains the zero function ($h_0 \equiv 0$). If a non-decreasing function $\Gamma : [0, \infty) \to [0, \infty)$, continuous at 0, satisfies the $\mathcal{H}$-consistency bound for all $h \in \mathcal{H}$ and any distribution:*

$$\mathcal{R}(h) - \mathcal{R}^*(\mathcal{H}) + \mathcal{M}(\mathcal{H}) \leq \Gamma(\mathcal{R}_\Phi(h) - \mathcal{R}_\Phi^*(\mathcal{H}) + \mathcal{M}_\Phi(\mathcal{H})),$$

*then, $\Gamma(t) \geq 1$ for all $t \geq 0$.*

*Proof Sketch.* The core obstruction is the decoupling of surrogate error and ranking error in the absence of a margin. We construct a *bad* sequence of hypotheses $\{h_k\}$ that converges to the zero function $h_0 \equiv 0$ in the surrogate space but remains maximally incorrect in the ranking space.

Specifically, consider a distribution concentrated on a single pair $y \succ y'$. For this concentrated distribution, the minimizability gaps vanish ($\mathcal{M}(\mathcal{H}) = \mathcal{M}_\Phi(\mathcal{H}) = 0$). By equicontinuity, we can find hypotheses that squash the score difference $h(x,y) - h(x,y')$ arbitrarily close to 0. Since $\Phi$ is continuous, the surrogate loss approaches its minimum $\Phi(0)$. However, because the sign is never strictly corrected (or is effectively random near 0), the 0-1 ranking error remains 1. Thus, $\mathcal{R}_\Phi \to \mathcal{R}_\Phi^*$ does not imply $\mathcal{R} \to \mathcal{R}^*$. A margin $\gamma$ is required to force the score difference away from this ambiguous region. $\square$

Theorem 3.1 establishes that for hypothesis sets typical of unconstrained neural networks, any $\mathcal{H}$-consistency bound is *vacuous*: even when minimizability gaps vanish, a small surrogate estimation error, $\mathcal{R}_\Phi(h) - \mathcal{R}_\Phi^*(\mathcal{H}) \approx 0$, only guarantees $\mathcal{R}(h) - \mathcal{R}^*(\mathcal{H}) \leq 1$, providing no meaningful guarantee that the true ranking error is small. This extends the inconsistency findings of Mao et al. (2023c, Theorem 2.1) to the equicontinuity properties inherent to modern LLMs.

## 4. $\mathcal{H}$-Consistency of Margin-Constrained Ranking

To resolve the inconsistency, we restrict our analysis to hypothesis sets $\mathcal{H}_\gamma$ that enforce a minimum separation margin $\gamma > 0$ between scores. Formally, we define the family of admissible hypothesis sets as: $\mathcal{H}_\gamma = \{\mathcal{H} \subseteq \mathcal{H}_{\text{all}} | \forall h \in \mathcal{H}, x, y \neq y' : |h(x,y) - h(x,y')| \geq \gamma, \exists h_+, h_- \in$

$\mathcal{H}$ s.t. $\forall x, y \neq y' : h_+(x,y) - h_+(x,y') = \gamma$ and $h_-(x,y) - h_-(x,y') = -\gamma\}$. This ensures every hypothesis separates pairs by at least $\gamma$, and the set contains hypotheses $h_+$ and $h_-$ achieving this boundary exactly.

Under this constraint, we can derive an $\mathcal{H}$-consistency bound between the surrogate loss and the target ranking loss for any hypothesis set $\mathcal{H} \in \mathcal{H}_\gamma$.

**Theorem 4.1** (**General $\mathcal{H}$-Consistency Bound**). *Let $\mathcal{H} \in \mathcal{H}_\gamma$ be a regular hypothesis set. Let $\Phi : \mathbb{R} \to \mathbb{R}_+$ be a convex, non-increasing function. For any $h \in \mathcal{H}$, the following $\mathcal{H}$-consistency bound holds:*

$$\mathcal{R}(h) - \mathcal{R}^*(\mathcal{H}) + \mathcal{M}(\mathcal{H})$$
$$\leq \frac{1}{\Phi(-\gamma) - \Phi(\gamma)}(\mathcal{R}_\Phi(h) - \mathcal{R}_\Phi^*(\mathcal{H}) + \mathcal{M}_\Phi(\mathcal{H}))$$

*Proof Sketch.* The proof relies on decomposing the conditional regret $\Delta\mathcal{C}_\mathcal{H}(h)$ into surrogate terms. The critical step is establishing a pointwise lower bound on the surrogate conditional error $\mathcal{C}_\Phi(h)$ for any hypothesis that misranks a pair (i.e., $\Delta h < 0$). Due to the margin constraint on $\mathcal{H}_\gamma$, any such misranking implies a score violation of at least $\gamma$ (i.e., $\Delta h \leq -\gamma$). By the convexity and monotonicity of $\Phi$, we show that the surrogate regret scales with the ranking regret: $\Delta\mathcal{C}_{\Phi,\mathcal{H}}(h) \geq (\Phi(-\gamma) - \Phi(\gamma))\Delta\mathcal{C}_\mathcal{H}(h)$. Rearranging this inequality yields the consistency coefficient. $\square$

Theorem 4.1 establishes that in realizable settings where minimizability gaps vanish, a surrogate estimation error of $\epsilon$ guarantees that the target estimation error $\mathcal{R}(h) - \mathcal{R}^*(\mathcal{H})$ is upper bounded by $\frac{\epsilon}{\Phi(-\gamma)-\Phi(\gamma)}$. The standard DPO formulation uses the logistic surrogate loss $\Phi_{\log}(u) = \log(1 + e^{-\beta u})$ for some temperature parameter $\beta > 0$. We can obtain the specific bound for DPO by evaluating the coefficient $\frac{1}{\Phi(-\gamma)-\Phi(\gamma)}$ explicitly. Substituting the logistic function: $\Phi_{\log}(-\gamma) - \Phi_{\log}(\gamma) = \log(e^{\beta\gamma}) = \beta\gamma$. Thus, for the specific case of DPO, the general coefficient simplifies to $\frac{1}{\beta\gamma}$.

**Corollary 4.2** (**$\mathcal{H}$-Consistency Bound for DPO**). *Let $\mathcal{H} \in \mathcal{H}_\gamma$ be a regular hypothesis set. For any $h \in \mathcal{H}$ and the logistic surrogate $\Phi_{\log} : u \mapsto \log(1 + e^{-\beta u})$, the following $\mathcal{H}$-consistency bound holds:*

$$\mathcal{R}(h) - \mathcal{R}^*(\mathcal{H}) + \mathcal{M}(\mathcal{H})$$
$$\leq \frac{1}{\beta\gamma}\Big(\mathcal{R}_{\Phi_{\log}}(h) - \mathcal{R}_{\Phi_{\log}}^*(\mathcal{H}) + \mathcal{M}_{\Phi_{\log}}(\mathcal{H})\Big)$$

The coefficient $\frac{1}{\beta\gamma}$ indicates that a larger margin $\gamma$ tightens the bound, implying that surrogate minimization is more effective at reducing the true ranking loss when the model maintains high confidence. However, identifying a hypothesis set that strictly satisfies this margin in practice is challenging. Similarly, a larger $\beta$ (lower temperature) reduces

the multiplicative factor, enhancing consistency, though excessively large $\beta$ values may degrade the optimization landscape of $\Phi_{\log}$ via vanishing gradients. Collectively, these results suggest that if an LLM (or reward model) maintains a minimum confidence gap $\gamma$ between response scores, minimizing the DPO objective serves as an effective proxy for minimizing the true ranking error.

**Topological Constraints and Non-Emptiness.** The class $\mathcal{H}_\gamma$ is technically non-empty for discrete domains or hypothesis sets with discontinuities, such as decision trees. Concrete instantiations include *Global Preference Models*, where functions $h(x,y) = s_y$ have fixed scores separated by $\gamma$; *Discontinuous Models* like decision trees that partition $\mathcal{X}$ into regions, allowing preferences to flip discontinuously at boundaries while satisfying local margins; and *Quantized Models*, such as neural networks with discrete output activations (e.g., $h(x,y) \in \{k\gamma : k \in \mathbb{Z}\}$) that ensure all distinct score differences are multiples of $\gamma$.

However, for continuous hypothesis sets such as neural networks on connected input domains $\mathcal{X}$, the strict margin condition $|h(x,y) - h(x,y')| \geq \gamma$ imposes a severe topological constraint. By the Intermediate Value Theorem, if the model were to change its preference between $y$ and $y'$ as the input $x$ varies, the score difference would necessarily pass through zero, violating the $\gamma$-condition. Consequently, Theorem 4.1 strictly applies only to models with fixed global preferences or discontinuous decision boundaries. While this establishes the theoretical *sufficiency* of margins for consistency, enforcing $\mathcal{H}_\gamma$ directly is too restrictive for practical deep learning. This limitation motivates our proposal of *Margin-Shifted Surrogates* in Section 5, which enforce margins via a soft penalty in the loss rather than a hard constraint on $\mathcal{H}$. This transition introduces a margin approximation gap $\mathcal{A}_\gamma(\mathcal{H})$ that quantifies the trade-off: we gain topological flexibility but incur a penalty proportional to the model's inability to fully satisfy the margin.

## 5. $\gamma$-Approximate $\mathcal{H}$-Consistency via Margin-Shifted Surrogates

Strict margin constraints ensure consistency but often render the hypothesis set non-convex. To maintain computational tractability, we propose the *margin-shifted surrogate loss*. Instead of constraining $\mathcal{H}$ directly, we shift the loss function to penalize correct classifications that fail to achieve a target confidence margin $\gamma > 0$.

**Definition 5.1** (Margin-Shifted Surrogate). Let $\Phi : \mathbb{R} \to \mathbb{R}_+$ be a convex, non-increasing surrogate function. For a target margin $\gamma > 0$, we define the margin-shifted surrogate loss $\mathsf{L}_{\Phi_\gamma}$ as:

$$\mathsf{L}_{\Phi_\gamma}(h, x, y, y', w) = \Phi\big(w \cdot (h(x,y) - h(x,y')) - \gamma\big). \quad (1)$$

## 5.1. $\gamma$-Approximate $\mathcal{H}$-Consistency Bound

Crucially, since the argument $u \mapsto u - \gamma$ is affine and $\Phi$ is convex, the composition $\mathsf{L}_{\Phi_\gamma}$ remains convex with respect to $h$. This allows for standard gradient-based optimization without the feasibility issues of non-convex constraints. We now derive a $\gamma$-approximate $\mathcal{H}$-consistency bound for this loss.

**Theorem 5.2** ($\gamma$-Shifted $\mathcal{H}$-Consistency Bound). *Let $\Phi: \mathbb{R} \to [0, +\infty)$ be non-increasing with $\Phi(-\gamma) > 0$ for some $\gamma > 0$. Define the shifted surrogate $\Phi_\gamma(u) = \Phi(u - \gamma)$. Then, for all $h \in \mathcal{H}$:*

$$\mathcal{R}(h) - \mathcal{R}^*(\mathcal{H}) + \mathcal{M}(\mathcal{H})$$
$$\leq \frac{1}{\Phi(-\gamma)}\big[\mathcal{R}_{\Phi_\gamma}(h) - \mathcal{R}^*_{\Phi_\gamma}(\mathcal{H}) + \mathcal{M}_{\Phi_\gamma}(\mathcal{H})\big] + \mathcal{A}_\gamma(\mathcal{H}),$$

*where $\mathcal{A}_\gamma(\mathcal{H}) = \frac{\mathbb{E}[\mathcal{C}^*_{\mathsf{L}_{\Phi_\gamma}}(\mathcal{H})]}{\Phi(-\gamma)} - \mathbb{E}[\mathcal{C}^*(\mathcal{H})]$ is the* margin approximation gap.

*Proof Sketch.* We use a calibration technique relating the 0-1 loss to the convex surrogate. The key insight is to upper-bound the discontinuous 0-1 indicator $1_{u \leq 0}$ using the shifted surrogate $\Phi(u - \gamma)$. By monotonicity of $\Phi$, if a pair is misranked ($u \leq 0$), then $u - \gamma \leq -\gamma$, implying $\Phi(u - \gamma) \geq \Phi(-\gamma)$. This allows us to establish the pointwise dominance: $1_{u \leq 0} \leq \frac{\Phi(u-\gamma)}{\Phi(-\gamma)}$. Taking expectations over the preference label $w$ yields a bound on the *conditional error* $\mathcal{C}(h)$ in terms of the shifted surrogate conditional error $\mathcal{C}_{\Phi_\gamma}(h)$. We then decompose the conditional regret $\Delta\mathcal{C}_\mathcal{H}$ into a scaled surrogate estimation term and the approximation gap $\mathcal{A}_\gamma$. This gap arises because the expected best-in-class surrogate conditional error $\mathbb{E}[\mathcal{C}^*_{\Phi_\gamma}(\mathcal{H})]$ (which enforces margins) is strictly higher than the unconstrained target error $\mathbb{E}[\mathcal{C}^*(\mathcal{H})]$ (which only requires correct signs). $\square$

The approximation gap $\mathcal{A}_\gamma(\mathcal{H})$ represents the price paid for requiring the model to satisfy the margin $\gamma$. Even if $\mathbb{E}[\mathcal{C}^*(\mathcal{H})] = 0$, $\mathcal{A}_\gamma(\mathcal{H})$ may be positive if the hypothesis set cannot produce scores with magnitude at least $\gamma$. However, as shown in Theorem E.3, for strictly separable finite data and scale-invariant models (like neural networks with unbounded logits), we can drive $\mathcal{A}_\gamma(\mathcal{H}) \to 0$ by scaling the outputs.

## 5.2. Properties of Shifted Consistency

**Proposition 5.3** (Vacuousness under Multiplicative Shift Invariance). *If a surrogate $\Phi$ satisfies $\Phi(u - \gamma) = C(\gamma)\Phi(u)$ for some $C(\gamma) > 0$, then for any $\gamma > 0$, the $\gamma$-shifted bound collapses to the unshifted bound ($\gamma = 0$). Specifically, the margin parameter cancels out, yielding: $\mathcal{R}(h) - \mathcal{R}^*(\mathcal{H}) + \mathcal{M}(\mathcal{H}) \leq \frac{1}{\Phi(0)}(\mathcal{R}_\Phi(h) - \mathcal{R}^*_\Phi(\mathcal{H}) + \mathcal{M}_\Phi(\mathcal{H})) + \mathcal{A}_0(\mathcal{H}).$*

The proof is presented in Appendix F. This highlights a critical failure mode: for the exponential loss ($\Phi(u) = e^{-u}$), shifting the margin scales the loss by $e^\gamma$, which cancels in the normalized bound. This explains why margin shifts are ineffective for purely exponential losses compared to logistic or hinge losses.

**Proposition 5.4** (Tightness of the $\gamma$-Shifted $\mathcal{H}$-Consistency Constant). *Let $\Phi$ be non-increasing with $\lim_{u \to \infty} \Phi(u) = 0$. The coefficient $C = \frac{1}{\Phi(-\gamma)}$ is optimal: for any $C' < C$, there exist a distribution $\mathcal{D}$ and hypothesis set $\mathcal{H}$ such that the consistency bound fails for some $h \in \mathcal{H}$: $\mathcal{R}(h) - \mathcal{R}^*(\mathcal{H}) + \mathcal{M}(\mathcal{H}) > C'\big[\mathcal{R}_{\Phi_\gamma}(h) - \mathcal{R}^*_{\Phi_\gamma}(\mathcal{H}) + \mathcal{M}_{\Phi_\gamma}(\mathcal{H})\big] + \mathcal{A}_\gamma(\mathcal{H}).$*

The proof is presented in Appendix G. This confirms that the trade-off between margin size and consistency is fundamental: enforcing a margin $\gamma$ necessarily incurs a factor of $1/\Phi(-\gamma)$, motivating surrogates like the Cubic Hinge (Section 6.3) that decay rapidly to minimize this cost.

## 5.3. Structure-Aware $\mathcal{H}$-Consistency Bounds

The previous analysis assumed a uniform margin parameter $\gamma$ across all inputs. However, in ranking tasks involving structured objects like text sequences, the difficulty of distinguishing a pair $(y, y')$ varies significantly based on their semantic similarity. Demanding a large margin for nearly identical responses introduces unnecessary bias, while a small global margin loosens the consistency guarantee for distinct pairs. To address this, we introduce a *pair-dependent margin function* $\Gamma: \mathcal{Y} \times \mathcal{Y} \to \mathbb{R}_+$. This allows us to enforce margins proportional to the dissimilarity of the responses, a technique akin to margin scaling in Structured SVMs (Tsochantaridis et al., 2005) and structured prediction in general (Mao et al., 2023f).

**Definition 5.5** (Structure-Aware Margin-Shifted Surrogate). Let $\Delta(y, y')$ be a non-negative distance metric between responses (e.g., normalized edit-distance or semantic embedding distance). We define the structure-aware margin as $\Gamma(y, y') = \tau\Delta(y, y')$ for a scaling factor $\tau > 0$. The corresponding surrogate loss is: $\mathsf{L}_{\Phi, \Gamma}(h, x, y, y', w) = \Phi(w \cdot (h(x, y) - h(x, y')) - \Gamma(y, y'))$.

**Algorithm: Structure-Aware DPO (SA-DPO).** By applying this framework to the standard DPO logistic loss, we obtain a novel objective we term SA-DPO. Substituting $\Phi_{\text{log}}$ and the implicit reward formulation, the objective becomes:

$$\mathcal{L}_{\text{SA-DPO}}(\pi_\theta) = -\mathbb{E}\big[\log \sigma(\beta w \cdot \Delta h_\theta(x) - \tau\Delta(y, y'))\big]$$
$$= -\mathbb{E}\bigg[\log \sigma\bigg(\beta w \log \frac{\pi_\theta(y|x)\pi_{\text{ref}}(y'|x)}{\pi_\theta(y'|x)\pi_{\text{ref}}(y|x)} - \tau\Delta(y, y')\bigg)\bigg].$$

Unlike standard DPO, which pushes for a constant log-probability gap regardless of content, SA-DPO dynamically

relaxes the margin constraint for a semantically similar pair $(y, y')$.

**Concrete Instantiations of** $\Gamma$. The choice of $\Delta(y, y')$ allows experts to inject prior knowledge into the alignment process: *Semantic Embedding Distance* defined as $\Delta(y, y') = 1 - \cos(E(y), E(y'))$ ensures the model learns strong discrimination only when responses are semantically distinct, preventing "margin collapse" on synonyms; *Edit Distance* is suitable for code generation or exact formatting tasks, where normalized edit distance ensures that small syntax errors are penalized less aggressively than complete hallucinations; and *Gold Reward Gap*, where if a teacher reward model $R^*$ is available, setting $\Gamma \propto |R^*(x, y) - R^*(x, y')|$ recovers a ranking-consistent formulation of Knowledge Distillation, forcing the student to respect the teacher's confidence gap rather than just the teacher's label.

The following result provides a guarantee for structure-aware algorithms.

**Theorem 5.6** (Structure-Aware $\gamma$-Shifted $\mathcal{H}$-Consistency Bound). *Let* $\Phi: \mathbb{R} \to [0, \infty)$ *be non-increasing with* $\Phi(0) > 0$, *and let* $\Gamma(y, y') > 0$. *Define the* inverse-margin weighted loss $\widetilde{\mathsf{L}}_{\Phi, \Gamma} = \frac{\mathsf{L}_{\Phi, \Gamma}}{\Phi(-\Gamma)}$. *Then, for any* $h \in \mathcal{H}$:

$$\mathcal{R}(h) - \mathcal{R}^*(\mathcal{H}) + \mathcal{M}(\mathcal{H})$$
$$\leq \left[ \mathcal{R}_{\widetilde{\mathsf{L}}_{\Phi, \Gamma}}(h) - \mathcal{R}^*_{\widetilde{\mathsf{L}}_{\Phi, \Gamma}}(\mathcal{H}) + \mathcal{M}_{\widetilde{\mathsf{L}}_{\Phi, \Gamma}}(\mathcal{H}) \right] + \mathcal{A}_{\Gamma}(\mathcal{H}),$$

*where* $\mathcal{A}_{\Gamma}(\mathcal{H}) = \mathbb{E}\left[\mathcal{C}^*_{\widetilde{\mathsf{L}}_{\Phi, \Gamma}}(\mathcal{H})\right] - \mathbb{E}\left[\mathcal{C}^*(\mathcal{H})\right]$ *is the structure-aware approximation gap.*

*Proof Sketch.* We extend the calibration technique from Theorem 5.2 by introducing a pair-dependent local margin $\gamma_{loc} = \Gamma(y, y')$. The key insight is that the inverse-margin weighted loss $\widetilde{\mathsf{L}}_{\Phi, \Gamma}$ acts as a normalized upper bound on the 0-1 loss: $1_{\Delta h \leq 0} \leq \frac{\Phi(\Delta h - \gamma_{loc})}{\Phi(-\gamma_{loc})} = \widetilde{\mathsf{L}}_{\Phi, \Gamma}$. Taking expectations allows us to bound the target error $\mathcal{R}(h)$ by the weighted surrogate error $\mathcal{R}_{\widetilde{\mathsf{L}}}(h)$, where the approximation gap $\mathcal{A}_{\Gamma}$ now captures the expected difficulty of satisfying these variable margins across the distribution. $\square$

**Benefit of Structure-Awareness.** The uniform bound (Theorem 5.2) is dominated by the worst-case margin pair, scaling with $1/\Phi(-\min_{y, y'} \Gamma(y, y'))$. If the distribution contains even one pair of very similar responses ($\min_{y, y'} \Gamma(y, y') \to 0$), the uniform coefficient explodes, rendering the bound loose. In contrast, the structure-aware bound depends on the *expectation* of the inverse margins. As long as the average pair is distinct, the consistency guarantee remains tight, even if hard pairs exist in the tail of the distribution. This justifies using weighted losses to focus optimization on pairs where the semantic difference warrants a strong ranking signal. Figure 1 illustrates this benefit.

## 6. Analysis of the Margin-Capacity Profile

The $\gamma$-shifted $\mathcal{H}$-consistency bound introduced in Theorem 5.2 presents a fundamental trade-off: increasing the margin $\gamma$ improves the consistency coefficient $\frac{1}{\Phi(-\gamma)}$, but potentially increases the surrogate risk if the hypothesis set $\mathcal{H}$ lacks the capacity to fully satisfy the margin.

To analyze this trade-off precisely, we revisit the pointwise inequality established in the proof of Theorem 5.2. For any hypothesis $h$, the true ranking error is directly upper-bounded by the normalized shifted surrogate error:

$$\mathcal{R}(h) \leq \frac{\mathcal{R}_{\Phi_{\gamma}}(h)}{\Phi(-\gamma)}. \tag{2}$$

This simplified bound highlights that the guarantee is governed by the ratio of the achieved loss to the margin penalty. In the practical regime where models have bounded outputs (e.g., logits constrained by normalization or fixed architectures), the numerator $\mathcal{R}_{\Phi_{\gamma}}(h)$ cannot be driven to zero if $\gamma$ exceeds the model's maximum output scale.

### 6.1. Analysis of the Margin Approximation Gap $\mathcal{A}_{\gamma}(\mathcal{H})$

The approximation gap term $\mathcal{A}_{\gamma}(\mathcal{H})$ defined in Theorem 5.2 acts as a *margin penalty*. It measures the discrepancy between the expected best possible conditional error under the strict margin requirement (surrogate conditional error) and the expected best possible conditional ranking error (target conditional error). We analyze its behavior in two key regimes: unbounded capacity (where the gap vanishes) and bounded capacity (where the gap creates an irreducible error floor). Both proofs are included in Appendix I.

**Proposition 6.1** (Vanishing Gap under Infinite Capacity). *Let* $\Phi: \mathbb{R} \to [0, \infty)$ *satisfy* $\Phi(-\gamma) > 0$ *and* $\lim_{u \to +\infty} \Phi(u) = 0$. *Let* $\mathcal{H}$ *be a hypothesis set closed under positive scalar multiplication (i.e.,* $h \in \mathcal{H} \implies \alpha h \in \mathcal{H}$ *for all* $\alpha > 0$). *If* $\mathcal{H}$ *is capable of perfect ranking on the support of* $\mathcal{D}$ *(i.e.,* $\mathcal{R}^*(\mathcal{H}) = 0$), *then for any finite margin* $\gamma > 0$:

$$\lim_{\alpha \to \infty} \mathcal{A}_{\gamma}(\mathcal{H}_{\alpha}) = 0, \tag{3}$$

*where* $\mathcal{H}_{\alpha}$ *denotes the scaled hypothesis set.*

**Implication for LLMs.** Proposition 6.1 explains why shifted DPO and similar methods work well with overparameterized neural networks. Since LLMs operate with unbounded logits, optimization can implicitly scale the weights ($\alpha \to \infty$) to satisfy any fixed margin $\gamma$. In this regime, $\mathcal{A}_{\gamma}$ is negligible and $\mathcal{H}$-consistency is governed solely by $\frac{1}{\Phi(-\gamma)}$.

**Proposition 6.2** (Penalty under Bounded Capacity). *Conversely, if the hypothesis set* $\mathcal{H}$ *has bounded outputs (e.g., due to the activation function used or strict spectral*

*normalization) such that* $\sup_{(h,x,y,y') \in \mathcal{H} \times \mathcal{X} \times \mathcal{Y} \times \mathcal{Y}} |h(x, y) - h(x, y')| \leq K$, *then for any margin* $\gamma > K$:

$$\mathcal{A}_\gamma(\mathcal{H}) \geq \frac{\Phi(K - \gamma)}{\Phi(-\gamma)} > 0, \qquad (4)$$

*even if the true ranking error* $\mathcal{R}^*(\mathcal{H})$ *is zero.*

**Empirical Estimation of the Approximation Gap.** To empirically estimate a proxy for this gap, we measured the *Margin Satisfaction Rate*—the percentage of test pairs where the model's learned score difference successfully exceeds the target theoretical margin. During our evaluation of the Qwen2.5-7B model (detailed in Section 7), Standard DPO satisfied a robust target margin proxy on only 58.0% of pairs, indicating a high practical approximation gap. SimPO improved this to 63.0% via its fixed margin. Strikingly, SA-DPO significantly increased overall margin satisfaction to 72.3%. This aligns perfectly with Proposition 6.2: in practical fine-tuning (especially with Low-Rank Adaptation (LoRA)), the model's maximum score difference (capacity $K$) is restricted. When a uniform margin exceeds this capacity ($\gamma > K$), the approximation gap becomes strictly positive. By dynamically scaling the margin $\gamma$ down for similar pairs while maintaining strict separation for distinct pairs, SA-DPO ensures the target margin remains within the model's capacity bounds, theoretically and practically shrinking the impact of $\mathcal{A}_\gamma(\mathcal{H})$.

## 6.2. The Margin-Capacity Profile

We introduce the *Margin-Capacity Profile* to quantify the efficiency of different loss functions in the bounded-capacity regime described above.

Consider a bounded hypothesis set $\mathcal{H}_K = \{h : \|h\|_\infty \leq K\}$. When we enforce a margin $\gamma > K$, the model cannot fully satisfy the margin constraint even on easy examples. The best achievable loss is limited by the capacity $K$.

**Definition 6.3** (Margin-Capacity Profile)**.** Let $\mathcal{H}_K$ be a hypothesis set with maximum score capacity $K$. The *Margin-Capacity Profile* of a surrogate $\Phi$ is the ratio of the best achievable loss at the capacity limit to the normalization factor:

$$\rho_\Phi(\gamma, K) = \frac{\Phi(K - \gamma)}{\Phi(-\gamma)}. \qquad (5)$$

A smaller $\rho_\Phi$ indicates that the loss function is more forgiving of capacity violations relative to the consistency coefficient it provides. This metric allows us to strictly order loss functions based on their tail behavior.

## 6.3. Heavier Tails: Polynomial Hinge Losses

We first compare the standard Logistic loss (used in DPO) against losses with heavier tails, specifically the Polynomial Hinge family.

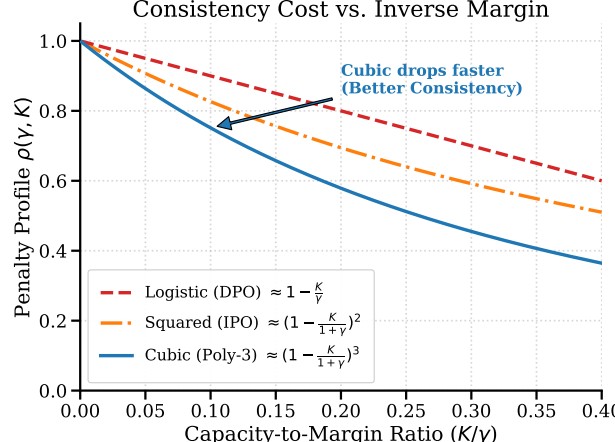

*Figure 2.* Theoretical Margin-Capacity Profiles $\rho$. The Logistic loss (DPO, red dashed) incurs a heavy penalty that decays linearly ($O(1/\gamma)$), meaning it struggles to guarantee consistency when margins are large. In contrast, the Cubic Hinge loss (Poly-3, blue solid) decays rapidly ($O(1/\gamma^3)$), indicating superior theoretical consistency for capacity-bounded models (verifying Proposition 6.5).

**Definition 6.4** (Polynomial Hinge Loss)**.** For a degree $k \geq 1$, the Polynomial Hinge Loss of degree $k$ is defined as:

$$\Phi_{\text{poly-}k}(u) := \max(0, 1 - u)^k. \qquad (6)$$

This family includes the standard Hinge loss ($k = 1$, linear tail similar to DPO) and the Squared Hinge loss ($k = 2$, quadratic tail similar to IPO (Azar et al., 2024)).

We can now derive the exact profile for these losses. For the Logistic loss $\Phi_{\log}(u) = \log(1 + e^{-\beta u})$, we analyze the regime where $\gamma > K$ are sufficiently large such that the loss is in its linear tail ($\Phi_{\log}(u) \approx -\beta u$).

**Proposition 6.5** (Monotonicity of Profile with Tail Degree)**.** *Let $\gamma > K > 0$. The Margin-Capacity Profile for the Polynomial Hinge Loss of degree $k$ is given by:*

$$\rho_k(\gamma, K) = \left(\frac{1 + \gamma - K}{1 + \gamma}\right)^k = \left(1 - \frac{K}{1 + \gamma}\right)^k. \qquad (7)$$

*For Logistic loss (linear tail), the profile is approximately:*

$$\rho_{\Phi_{\log}}(\gamma, K) \approx 1 - \frac{K}{\gamma}. \qquad (8)$$

*Proof Sketch.* We explicitly evaluate the profile ratio $\rho = \Phi(K - \gamma)/\Phi(-\gamma)$. For Polynomial Hinge losses, the specific algebraic form yields $\rho_k = \left(\frac{1+\gamma-K}{1+\gamma}\right)^k = \left(1 - \frac{K}{1+\gamma}\right)^k$. For the Logistic loss, we analyze the linear tail regime ($\Phi(u) \approx -\beta u$ for $u \ll 0$), which yields $\rho_{\log} \approx \frac{\beta(\gamma-K)}{\beta\gamma} = 1 - \frac{K}{\gamma}$. $\qquad\square$

Since the base term satisfies $0 < 1 - \frac{K}{1+\gamma} < 1$, the profile $\rho_k$ decreases exponentially with the degree $k$. This implies

a strict hierarchy of theoretical consistency for capacity-bounded models: $\rho_{\text{cubic}} < \rho_{\text{sq-hinge}} \approx \rho_{\text{IPO}} < \rho_{\text{logistic}}$. While DPO (Logistic) incurs a penalty that decays linearly with the margin gap, IPO (Squared) decays quadratically, and Cubic Hinge decays cubically. Consequently, losses with heavier tails allow for tighter consistency bounds when the margin $\gamma$ is pushed near or beyond the model's physical capacity $K$, as illustrated in Figure 2.

### 6.4. Bounded Tails: The Comp-Sum Family

Beyond the Polynomial Hinge family, we consider the broader class of *Comp-Sum losses* (Mao et al., 2023e), which includes the Generalized Cross Entropy (GCE) and Mean Absolute Error (MAE). These losses are often preferred in classification for their robustness to label noise.

For ranking, the GCE loss with parameter $q \in (0, 1]$ is defined via the mapping $\Phi_{\text{GCE}}(u) = \frac{1 - \sigma(u)^q}{q}$. Unlike the Logistic or Hinge losses, $\Phi_{\text{GCE}}$ is *bounded* as the score difference approaches negative infinity: $\lim_{u \to -\infty} \Phi_{\text{GCE}}(u) = \frac{1}{q}$.

**Proposition 6.6** (Profile of Bounded Losses). *For any bounded surrogate loss where $\lim_{u \to -\infty} \Phi(u) = C > 0$, the Margin-Capacity Profile satisfies:* $\lim_{\gamma \to \infty} \rho_{\Phi}(\gamma, K) = 1$.

The proof is presented in Appendix J. This reveals a fundamental limitation of bounded losses in the margin-shifted framework. Unlike Polynomial Hinge losses where the profile decays to zero (allowing the approximation gap to vanish for large margins), bounded losses maintain a constant penalty ratio $\rho \approx 1$. This suggests that while GCE may be robust to label noise, it provides weaker consistency guarantees for capacity-bounded models compared to heavy-tailed unbounded losses like the Cubic Hinge.

### 6.5. Theoretically Optimal Margin Parameter

The analysis above suggests that for heavy-tailed losses, we can afford larger margins. However, for a fixed loss like Logistic (DPO), we must optimize $\gamma$ to balance the coefficient against the capacity penalty.

Let $K$ be the maximum score capacity of the hypothesis set. Assuming the model is capacity-limited ($\mathcal{R}_{\Phi_{\gamma}}^{*}(\mathcal{H}) > 0$), the bound $B(\gamma)$ can be written as:

$$B(\gamma) = \underbrace{\frac{1}{\Phi(-\gamma)}\epsilon}_{\text{Estimation Term}} + \underbrace{\left(\frac{\Phi(K - \gamma)}{\Phi(-\gamma)} - 0\right)}_{\text{Approximation Term}}, \quad (9)$$

where $\epsilon = \mathcal{R}_{\Phi_{\gamma}}(h) - \mathcal{R}_{\Phi_{\gamma}}^{*}(\mathcal{H})$ is the estimation error.

**Proposition 6.7** (Optimal Margin for Bounded Models). *For $\Phi_{\log}(u) = \log(1 + e^{-\beta u})$ and a hypothesis set with score capacity $K$, the optimal margin $\gamma^{*}$ that minimizes the*

$\gamma$-*shifted $\mathcal{H}$-consistency bound is the solution to:*

$$\gamma^{*} \approx K + \frac{1}{\beta}\log\left(\frac{\epsilon}{\beta K}\right). \quad (10)$$

The proof is presented in Appendix L. This result suggests the optimal margin $\gamma^{*}$ is slightly larger or smaller than the model capacity $K$, depending on the estimation error $\epsilon$. In practice, since modern LLMs have very large $K$ (unbounded logits), the optimal strategy is to set $\gamma$ as large as optimization stability permits.

## 7. Experiments

To empirically validate our theoretical findings, we conduct two controlled experiments isolating optimization and capacity phenomena, followed by real-world evaluations on the UltraFeedback (Cui et al., 2024) and Argilla DPO-Mix-7k (Argilla, 2023) benchmarks.

### 7.1. Implementation Overview

We evaluate using the Llama-3-8B base model (Dubey et al., 2024) accelerated via Unsloth (Daniel Han & team, 2023) and TRL (von Werra et al., 2020), as well as the Qwen2.5-7B-Instruct architecture (Yang et al., 2024). All models are fine-tuned using Low-Rank Adaptation (LoRA) (Hu et al., 2022). We provide the complete set of hyperparameters, including learning rates, batch sizes, margin configurations ($\tau, \gamma$), and embedding model details in Appendix M.

### 7.2. Controlled Validation

We first investigate two key theoretical predictions: the instability of uniform margins on synonyms and the impact of loss tail heaviness on capacity-constrained models.

**Synonym Stress Test.** We constructed a synthetic dataset of 100 semantically identical (Levenshtein dist < 0.1) but lexically distinct pairs to test if uniform margins induce instability. As shown in Figure 3 (Left), Standard DPO struggles to converge, stalling at a high loss ($0.1695 \pm 0.003$ over 5 seeds) as it effectively "hallucinates" a preference to satisfy the margin. In contrast, SA-DPO dynamically relaxes the margin for these pairs, achieving stable convergence with near-zero loss ($0.003 \pm 0.001$), as detailed in Table 1.

**Margin-Capacity Profile.** We validated the hierarchy of loss functions derived in Section 6 using the Anthropic HH-RLHF dataset (Bai et al., 2022) with a hard margin of $\gamma = 1.0$. Figure 3 (Right) confirms the theoretical prediction that consistency is governed by tail heaviness. As reported in Table 2, DPO (Linear tail) fails to satisfy the margin, stalling at $71.6\% \pm 1.1\%$ accuracy (over 5 seeds). IPO (Quadratic tail) improves this to $94.5\% \pm 0.8\%$ but

*Table 1.* Optimization Performance on Synonyms. Final training metrics on the Synonym Stress Test (mean ± std. dev. over 5 runs). Standard DPO stalls at a non-trivial loss, while SA-DPO achieves perfect convergence.

| Method | Final Loss | Implicit Margin | Status |
|---|---|---|---|
| Standard DPO | $0.1695 \pm 0.003$ | $\approx 1.70$ | Stalled |
| **SA-DPO (Ours)** | $0.003 \pm 0.001$ | $\approx 6.32$ | **Converged** |

*Table 2.* Final Ranking Accuracy (Margin-Capacity). Performance with standard LoRA rank $r = 8$ and $\gamma = 1.0$ (mean ± std. dev. over 5 runs). The hierarchy of consistency follows the heaviness of the loss tail.

| Metric | DPO (Linear) | IPO (Quadratic) | Poly-3 (Cubic) |
|---|---|---|---|
| Final Accuracy | $71.6\% \pm 1.1\%$ | $94.5\% \pm 0.8\%$ | $99.7\% \pm 0.3\%$ |

converges slowly, while the Cubic Hinge (Poly-3) rapidly achieves near-perfect consistency ($99.7\% \pm 0.3\%$) by generating larger gradients for margin violations.

### 7.3. Real-World Evaluation

To analyze the impact of adaptivity on heterogeneous data, we evaluate Ranking Accuracy (RA) on distinct vs. ambiguous test splits (separated by the median semantic distance).

**UltraFeedback (Llama-3-8B).** Results in Table 3 (reported as mean ± standard deviation over five independent runs) show that SA-DPO outperforms baselines across all splits. Crucially, on the *Hard Ambiguous* subset (top 20% most ambiguous), SA-DPO demonstrates substantial gains over SimPO (0.700 vs. 0.650). This confirms that scaling the margin constraint by semantic distance prevents overfitting on ambiguous pairs while maintaining robustness on distinct ones.

**Argilla DPO-Mix-7k (Qwen2.5-7B).** To demonstrate architectural and distributional robustness, we additionally evaluated DPO, SimPO, and SA-DPO on the Argilla DPO-Mix-7k dataset (Argilla, 2023) using the Qwen2.5-7B-Instruct model (Yang et al., 2024) across 5 random seeds. SA-DPO achieved a Distinct RA of $0.798 \pm 0.006$ and an Ambiguous RA of $0.746 \pm 0.006$, consistently outperforming both DPO (Distinct: $0.774 \pm 0.002$ / Ambiguous: $0.727 \pm 0.005$) and SimPO (Distinct: $0.787 \pm 0.005$ / Ambiguous: $0.733 \pm 0.004$).

**Downstream Generation Quality.** While ranking accuracy directly measures the consistency of the optimization objective, downstream generation quality is critical for practical validation. We generated open-ended responses from our fine-tuned Qwen2.5-7B models (Yang et al., 2024) using standard MT-Bench (Zheng et al., 2023) style conversational prompts. Evaluated head-to-head using the widely adopted PairRM (LLM-Blender) cross-encoder framework (Jiang et al., 2023), SA-DPO achieved a robust win-rate of 58.5%

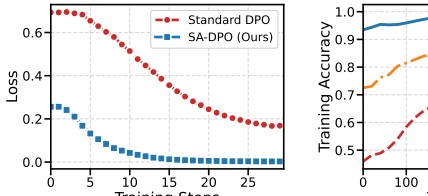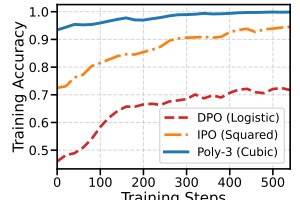

*Figure 3.* Controlled Validation. (Left) Synonym Stability: Standard DPO (dashed) stalls by enforcing margins on identical pairs, while SA-DPO (solid) adapts to achieve smooth convergence. (Right) Margin-Capacity Profile: DPO (Linear) fails to satisfy the margin. IPO (Quadratic) converges slowly, while Poly-3 (Cubic) rapidly achieves near-perfect consistency, confirming that heavier tails drive efficient constraint satisfaction.

*Table 3.* UltraFeedback Results. Ranking Accuracy (RA) on test splits (mean ± std. dev. over 5 runs). SA-DPO gains are most significant on the *Hard Ambiguous* subset.

| Metric | DPO | SimPO | SA-DPO |
|---|---|---|---|
| RA (Distinct) | $0.766 \pm 0.005$ | $0.780 \pm 0.003$ | **$0.790 \pm 0.004$** |
| RA (Ambiguous) | $0.716 \pm 0.005$ | $0.722 \pm 0.003$ | **$0.734 \pm 0.004$** |
| RA (Hard Subset) | $0.590 \pm 0.015$ | $0.650 \pm 0.013$ | **$0.700 \pm 0.003$** |

against Standard DPO. This confirms that correctly relaxing the margin on synonymous pairs prevents the model from learning hallucinated conversational artifacts, translating theoretical consistency improvements into demonstrably better generation quality.

## 8. Conclusion

We have shown that the surrogate losses underlying DPO and related alignment methods are *inconsistent* for the equicontinuous hypothesis sets characteristic of neural networks: minimizing these losses provides no guarantee that the true ranking error decreases. This is a fundamental limitation of the most widely deployed alignment paradigm. We resolve it by proving that enforcing a confidence margin $\gamma$ is both necessary and sufficient for $\mathcal{H}$-consistency, and introduce SA-DPO, which adapts this margin to the semantic distance between responses to avoid instability on near-synonymous pairs. Our analysis of the Margin-Capacity Profile further reveals a strict hierarchy: heavy-tailed losses (IPO, Cubic Hinge) offer provably superior consistency for capacity-bounded models compared to the logistic loss of DPO. These theoretical findings are validated empirically: SA-DPO consistently outperforms both DPO and SimPO on ranking accuracy across datasets and architectures, and achieves a 58.5% win-rate in downstream generation quality. We also unify Bregman-regularized RLHF methods under this framework (Appendix N). Future directions include extending these bounds to listwise ranking, online exploration, and non-transitive preference models.

## Acknowledgments

We thank our colleague Jon Schneider for discussions about a preliminary version of this paper.

## Impact Statement

This paper presents work whose goal is to advance the field of Machine Learning. There are many potential societal consequences of our work, none which we feel must be specifically highlighted here.

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

# Contents of Appendix

# A. Extended Related Work

Our work intersects with three distinct bodies of literature: direct preference alignment, learning theory for ranking, and structured prediction.

**Direct Alignment from Preferences.**    The paradigm of aligning LLMs via preference data was popularized by RLHF (Christiano et al., 2017; Stiennon et al., 2020), which involves learning an explicit reward model. Recently, implicit approaches have gained prominence. Rafailov et al. (2023) introduced Direct Preference Optimization (DPO), deriving the policy update directly from the optimal reward condition. Subsequent works have proposed variations to address DPO's limitations: Azar et al. (2024) introduced IPO to prevent over-fitting by regularizing the reward gap, which we analyze as a margin-shifted squared loss. Zhao et al. (2023) proposed SLiC-HF, effectively using a hinge loss on the probability ratios, which aligns with our analysis of linear-tail margins. More recently, Meng et al. (2024) empirically demonstrated the benefits of incorporating a hard margin term into the DPO objective (SimPO) and removing the reference model. Our theory provides a rigorous justification for such modifications, proving them necessary for consistency. Other non-pairwise approaches like KTO (Ethayarajh et al., 2024) model utility directly, offering an alternative to the ranking formulation we study here. While recent works have extended preference optimization to the online setting to address exploration and distribution shift (Xiong et al., 2024; Guo et al., 2024), our analysis focuses on the consistency of the optimization objective itself. Since online methods typically optimize a surrogate loss in their inner loop, our findings on margin-consistency remain relevant in iterative settings.

**Consistency of Ranking.**    The consistency of surrogate losses is a classical problem in statistical learning theory. Bartlett et al. (2006) provided the foundational characterization of convex surrogates consistent with the 0-1 binary classification loss. For the ranking setting, Duchi et al. (2010) and Calauzenes et al. (2012) analyzed consistency, highlighting the difficulty of achieving it without strong domain assumptions. However, these classical results typically assume the hypothesis space is the set of all measurable functions ($\mathcal{H}_{all}$). In deep learning tasks, $\mathcal{H}$ is restricted (e.g., to a family of neural networks), rendering these universal guarantees inapplicable. Our analysis builds upon the framework of $\mathcal{H}$-*consistency* developed by Awasthi, Mao, Mohri, and Zhong (2022a;b) and recently extended by Mao et al. (2023e;c;a;b;d; 2024a;b;c;d;e;f;g; 2025c;a;b); Mohri et al. (2024); Cortes et al. (2024; 2025a;b; 2026a;b); Mao (2025); Zhong (2025); DeSalvo et al. (2025); Montreuil et al. (2025b;a; 2026e;b;d;a;c;f;g); Mohri & Zhong (2026c;b;a); Mohri et al. (2026). We show that standard $\mathcal{H}$-consistency fails for equicontinuous hypothesis sets typical of neural networks. Instead, we propose the notion of $\gamma$-*shifted $\mathcal{H}$-consistency* and prove that this guarantee holds for the margin-shifted versions of DPO and IPO used in the LLM setting.

**Margins and Structured Prediction.**    The concept of enforcing margins for consistency draws on the rich history of Support Vector Machines (SVMs) (Cortes & Vapnik, 1995) and their extension to ranking (Herbrich et al., 2000; Joachims, 2002). Our proposal for *Structure-Aware* margins is inspired by Structured SVMs (Tsochantaridis et al., 2005) and structured prediction (Mao, Mohri, and Zhong, 2023f), as well as cost-sensitive classification (Zadrozny et al., 2003), which scale penalties based on the severity of the error. In the context of LLMs, this connects to *margin-aware* fine-tuning methods that have shown empirical promise (Yuan et al., 2023; Zhou et al., 2023) but lacked a unifying theoretical consistency proof until now.

**Social Choice and Game-Theoretic Perspectives.**    Recent theoretical works have also critically examined DPO from perspectives beyond statistical consistency. Gölz et al. (2026) use social choice theory to show that DPO suffers from high distortion in aggregating diverse human preferences, effectively acting as a Borda count. While our work focuses on statistical consistency under a ground-truth assumption, our structure-aware framework offers a potential mechanism to mitigate social distortion by assigning lower margins to controversial pairs. Similarly, Munos et al. (2024) challenge the scalar reward assumption of the Bradley-Terry model, proposing Nash Learning from Human Feedback (NLHF) to handle non-transitive preferences via game-theoretic equilibria. Our work complements these approaches by focusing on the optimization landscape: we address the inconsistency of the learning objective itself, rather than the aggregation rule or the underlying preference model.

# B. Proof of Theorem 3.1

**Theorem 3.1** (**Negative Results for Equicontinuous** $\mathcal{H}$). *Assume that $\mathcal{H}$ is regular, equicontinuous, and contains the zero function ($h_0 \equiv 0$). If a non-decreasing function $\Gamma : [0, \infty) \to [0, \infty)$, continuous at 0, satisfies the $\mathcal{H}$-consistency bound for*

*all $h \in \mathcal{H}$ and any distribution:*

$$\mathcal{R}(h) - \mathcal{R}^*(\mathcal{H}) + \mathcal{M}(\mathcal{H}) \leq \Gamma(\mathcal{R}_\Phi(h) - \mathcal{R}_\Phi^*(\mathcal{H}) + \mathcal{M}_\Phi(\mathcal{H})),$$

*then, $\Gamma(t) \geq 1$ for all $t \geq 0$.*

*Proof.* Consider a distribution $\mathcal{D}$ that is concentrated entirely on a single tuple $(x, y, y', w)$ with deterministic label $w = -1$ (implying $y' > y$). The true optimal error is $\mathcal{R}^*(\mathcal{H}) = \inf_{h \in \mathcal{H}} 1_{w \neq \text{sign}(h(x,y) - h(x,y'))}$. Since $\mathcal{H}$ is regular, there exists $h \in \mathcal{H}$ such that $-1 = \text{sign}(h(x,y) - h(x,y'))$, implying $\mathcal{R}^*(\mathcal{H}) = 0$. Let $h_0$ be the zero function ($h_0(x, y) = 0$). Then $\mathcal{R}(h_0) = 1_{-1 \neq 1} = 1$. By the *equicontinuity* of $\mathcal{H}$, for any $\epsilon > 0$, we can choose the support elements $x, y \neq y'$ such that $|h(x,y) - h(x,y')| < \epsilon$ for all $h \in \mathcal{H}$.

The surrogate error for any $h \in \mathcal{H}$ is $\mathcal{R}_\Phi(h) = \Phi(-1 \cdot (h(x,y) - h(x,y')))$. Then $\mathcal{R}_\Phi(h_0) = \Phi(0)$. Since the function $\Phi$ is non-increasing, the range of possible surrogate errors is bounded:

$$\mathcal{R}_\Phi(h) \in [\Phi(\epsilon), \Phi(-\epsilon)]$$

Therefore, the optimal surrogate error is lower-bounded by $\mathcal{R}_\Phi^*(\mathcal{H}) \geq \Phi(\epsilon)$. Applying the assumed $\mathcal{H}$-consistency bound to the zero function $h_0$, we get:

$$\mathcal{R}(h_0) - \mathcal{R}^*(\mathcal{H}) + \mathcal{M}(\mathcal{H}) \leq \Gamma(\mathcal{R}_\Phi(h_0) - \mathcal{R}_\Phi^*(\mathcal{H}) + \mathcal{M}_\Phi(\mathcal{H})),$$

where $\mathcal{M}(\mathcal{H}) = \mathcal{M}_\Phi(\mathcal{H}) = 0$ for this distribution concentrated on a single tuple. Substituting the known values and lower bound:

$$1 - 0 \leq \Gamma(\Phi(0) - \mathcal{R}_\Phi^*(\mathcal{H})) \leq \Gamma(\Phi(0) - \Phi(\epsilon))$$

Since $\Phi$ is continuous, taking the limit as $\epsilon \to 0$ yields $\Phi(\epsilon) \to \Phi(0)$. Given that $\Gamma$ is continuous at 0 and non-decreasing, we conclude that:

$$\Gamma(0) \geq 1$$

By the non-decreasing nature of $\Gamma$, this implies $\Gamma(t) \geq 1$ for all $t \geq 0$. $\qquad\square$

## C. Proof of Theorem 4.1

**Theorem 4.1** (**General $\mathcal{H}$-Consistency Bound**). *Let $\mathcal{H} \in \mathcal{H}_\gamma$ be a regular hypothesis set. Let $\Phi : \mathbb{R} \to \mathbb{R}_+$ be a convex, non-increasing function. For any $h \in \mathcal{H}$, the following $\mathcal{H}$-consistency bound holds:*

$$\mathcal{R}(h) - \mathcal{R}^*(\mathcal{H}) + \mathcal{M}(\mathcal{H}) \leq \frac{1}{\Phi(-\gamma) - \Phi(\gamma)}(\mathcal{R}_\Phi(h) - \mathcal{R}_\Phi^*(\mathcal{H}) + \mathcal{M}_\Phi(\mathcal{H}))$$

*Proof.* Fix a tuple $(x, y, y')$ and let $\eta = \eta(x, y, y')$. The conditional error for the target loss is $\mathcal{C}(h) = \eta 1_{\Delta h < 0} + (1 - \eta) 1_{\Delta h \geq 0}$. Since $\mathcal{H}$ is regular, the best-in-class conditional error is $\mathcal{C}^*(\mathcal{H}) = \min\{\eta, 1 - \eta\}$.

Consider the set of hypotheses $\overline{\mathcal{H}}(x, y, y')$ that misclassify the pair relative to the Bayes optimal decision. For any $h \in \overline{\mathcal{H}}(x, y, y')$, the conditional regret is:

$$\Delta \mathcal{C}_{\mathcal{H}}(h) = \mathcal{C}(h) - \mathcal{C}^*(\mathcal{H}) = |2\eta - 1|.$$

For the surrogate loss, the conditional error is $\mathcal{C}_\Phi(h) = \eta \Phi(\Delta h) + (1 - \eta) \Phi(-\Delta h)$. By the margin assumption, $\mathcal{H}$ contains hypotheses $h_+, h_-$ achieving score differences $\Delta h = \pm \gamma$. The best-in-class surrogate conditional error is upper bounded by the minimum error among these:

$$\mathcal{C}_\Phi^*(\mathcal{H}) \leq \min\{\mathcal{C}_\Phi(h_+), \mathcal{C}_\Phi(h_-)\}$$
$$= \max\{\eta, 1 - \eta\}\Phi(\gamma) + \min\{\eta, 1 - \eta\}\Phi(-\gamma).$$

Now, consider a misclassifying hypothesis $h \in \overline{\mathcal{H}}(x, y, y')$. Without loss of generality, assume $\eta \geq 1/2$. For $h$ to misclassify, we must have $\Delta h < 0$. Furthermore, the $\gamma$-margin condition on $\mathcal{H}$ implies that the magnitude of this violation is at least $\gamma$, i.e., $\Delta h \leq -\gamma$.

Let $g(u) \coloneqq \eta\Phi(u) + (1-\eta)\Phi(-u)$ be the surrogate error as a function of the score difference $u = \Delta h$. We show that $g(u)$ is non-increasing for $u < 0$. Consider scores $u_1 < u_2 \le -\gamma < 0$. Let $S_1, S_2$ be the secant slopes of $\Phi$ on $[u_1, u_2]$ and $[-u_2, -u_1]$ respectively. Since $\Phi$ is convex and non-increasing, we have $S_1 \le S_2 \le 0$. The change in error is:

$$g(u_1) - g(u_2) = \eta[\Phi(u_1) - \Phi(u_2)] + (1-\eta)[\Phi(-u_1) - \Phi(-u_2)]$$
$$= (u_2 - u_1)[(1-\eta)S_2 - \eta S_1] \ge 0,$$

where the inequality holds because $\eta \ge 1 - \eta$ and $-S_1 \ge -S_2 \ge 0$. Thus, $g(u)$ is minimized at the boundary $u = -\gamma$:

$$\mathcal{C}_\Phi(h) \ge g(-\gamma) = \eta\Phi(-\gamma) + (1-\eta)\Phi(\gamma).$$

Subtracting the upper bound for $\mathcal{C}^*_\Phi(\mathcal{H})$ from this lower bound yields the surrogate conditional regret:

$$\Delta\mathcal{C}_{\Phi,\mathcal{H}}(h) = \mathcal{C}_\Phi(h) - \mathcal{C}^*_\Phi(\mathcal{H})$$
$$\ge [\eta\Phi(-\gamma) + (1-\eta)\Phi(\gamma)] - [\eta\Phi(\gamma) + (1-\eta)\Phi(-\gamma)]$$
$$= (2\eta - 1)[\Phi(-\gamma) - \Phi(\gamma)]$$
$$= (\Phi(-\gamma) - \Phi(\gamma))\Delta\mathcal{C}_\mathcal{H}(h).$$

Taking the expectation over $\mathcal{D}$ yields the stated bound. $\qquad\square$

## D. Proof of Theorem 5.2

**Theorem 5.2** ($\gamma$-Shifted $\mathcal{H}$-Consistency Bound). *Let* $\Phi\colon\mathbb{R} \to [0, +\infty)$ *be non-increasing with* $\Phi(-\gamma) > 0$ *for some* $\gamma > 0$. *Define the shifted surrogate* $\Phi_\gamma(u) = \Phi(u - \gamma)$. *Then, for all* $h \in \mathcal{H}$:

$$\mathcal{R}(h) - \mathcal{R}^*(\mathcal{H}) + \mathcal{M}(\mathcal{H}) \le \frac{1}{\Phi(-\gamma)}[\mathcal{R}_{\Phi_\gamma}(h) - \mathcal{R}^*_{\Phi_\gamma}(\mathcal{H}) + \mathcal{M}_{\Phi_\gamma}(\mathcal{H})] + \mathcal{A}_\gamma(\mathcal{H}),$$

*where* $\mathcal{A}_\gamma(\mathcal{H}) = \frac{\mathbb{E}[\mathcal{C}^*_{\mathsf{L}_{\Phi_\gamma}}(\mathcal{H})]}{\Phi(-\gamma)} - \mathbb{E}[\mathcal{C}^*(\mathcal{H})]$ *is the* margin approximation gap.

*Proof.* Fix $h \in \mathcal{H}$. For any tuple $(x, y, y', w)$, let $\Delta h(x) \coloneqq w(h(x, y) - h(x, y'))$. The 0-1 pairwise ranking loss is upper bounded by $1_{\Delta h(x) \le 0}$. Using the definition $\Phi_\gamma(u) = \Phi(u - \gamma)$ and the monotonicity of $\Phi$:

$$u \le 0 \implies u - \gamma \le -\gamma \implies \Phi_\gamma(u) = \Phi(u - \gamma) \ge \Phi(-\gamma) > 0.$$

Thus, we have the pointwise inequality $1_{u \le 0} \le \frac{\Phi_\gamma(u)}{\Phi(-\gamma)}$. Applying this with $u = \Delta h(x)$:

$$\mathsf{L}_{0-1}(h, x, y, y', w) \le \frac{\mathsf{L}_{\Phi_\gamma}(h, x, y, y', w)}{\Phi(-\gamma)}.$$

Taking expectations over $w$ yields $\mathcal{C}(h) \le \frac{\mathcal{C}_{\mathsf{L}_{\Phi_\gamma}}(h)}{\Phi(-\gamma)}$ (where we suppress the arguments $x, y, y'$ for brevity). Since this holds for all $h \in \mathcal{H}$, it holds for the infimum:

$$\mathcal{C}^*(\mathcal{H}) \le \inf_{h \in \mathcal{H}} \frac{\mathcal{C}_{\mathsf{L}_{\Phi_\gamma}}(h)}{\Phi(-\gamma)} = \frac{\mathcal{C}^*_{\mathsf{L}_{\Phi_\gamma}}(\mathcal{H})}{\Phi(-\gamma)}.$$

This implies the approximation gap is non-negative: $\mathcal{A}_\gamma(\mathcal{H}) \ge 0$. We now decompose the conditional regret:

$$\Delta\mathcal{C}_\mathcal{H}(h) = \mathcal{C}(h) - \mathcal{C}^*(\mathcal{H})$$
$$\le \frac{\mathcal{C}_{\mathsf{L}_{\Phi_\gamma}}(h)}{\Phi(-\gamma)} - \mathcal{C}^*(\mathcal{H})$$
$$= \frac{1}{\Phi(-\gamma)}[\mathcal{C}_{\mathsf{L}_{\Phi_\gamma}}(h) - \mathcal{C}^*_{\mathsf{L}_{\Phi_\gamma}}(\mathcal{H})] + \left[\frac{\mathcal{C}^*_{\mathsf{L}_{\Phi_\gamma}}(\mathcal{H})}{\Phi(-\gamma)} - \mathcal{C}^*(\mathcal{H})\right]$$
$$= \frac{1}{\Phi(-\gamma)}\Delta\mathcal{C}_{\mathsf{L}_{\Phi_\gamma},\mathcal{H}}(h) + \left[\frac{\mathcal{C}^*_{\mathsf{L}_{\Phi_\gamma}}(\mathcal{H})}{\Phi(-\gamma)} - \mathcal{C}^*(\mathcal{H})\right].$$

Taking the expectation over $(x, y, y')$ yields the final result:

$$\mathcal{R}(h) - \mathcal{R}^*(\mathcal{H}) + \mathcal{M}(\mathcal{H}) \leq \frac{1}{\Phi(-\gamma)}\big[\mathcal{R}_{\Phi_\gamma}(h) - \mathcal{R}^*_{\Phi_\gamma}(\mathcal{H}) + \mathcal{M}_{\Phi_\gamma}(\mathcal{H})\big] + \mathcal{A}_\gamma(\mathcal{H}).$$

$\square$

## E. Consistency on Finite Domains

While Theorem 3.1 establishes inconsistency for general equicontinuous hypothesis sets, the specific domain of Large Language Models offers a structural advantage: the input space consists of sequences over a finite vocabulary. In practice, the support of the preference distribution $\mathcal{D}$ is restricted to a finite set $S \subset \mathcal{X} \times \mathcal{Y} \times \mathcal{Y}$.

We can show that under the assumption of finite support and model realizability, consistency is guaranteed because the hypothesis set of over-parameterized neural networks can achieve an arbitrarily large separation margin.

**Definition E.1** (Finite Realizable Support). The support $S = \text{supp}(\mathcal{D})$ is finite. Furthermore, the distribution is realizable on $S$: for any $(x, y, y') \in S$ with $y \neq y'$, the preference label $w$ is deterministic ($|\eta(x, y, y') - 1/2| = 1/2$) and non-contradictory.

**Definition E.2** (Strict Separability via Logit Scaling). We assume the hypothesis set $\mathcal{H}$ is parameterized by logits $z_\theta(x, \cdot)$ such that the policy is $\pi_\theta(y|x) \propto \exp(z_\theta(x, y))$. We say $\mathcal{H}$ is *strictly separable* on $S$ if there exists a parameter $\theta$ such that for all $(x, y, y', w) \in S$:

$$z_\theta(x, y_{\text{win}}) > z_\theta(x, y_{\text{lose}}),$$

where $y_{\text{win}}$ is the preferred response. Furthermore, we assume $\mathcal{H}$ is closed under positive scalar multiplication of the logits ($z \to \alpha z$ for $\alpha > 0$).

**Theorem E.3** (Consistency via Margin Scaling). *Under the assumptions of Finite Realizable Support and Strict Separability, minimizing the DPO logistic surrogate loss is consistent with respect to the 0-1 ranking loss. Specifically, for any error tolerance $\epsilon > 0$, there exists a hypothesis $h \in \mathcal{H}$ (achieved by scaling logits) such that the $\mathcal{H}$-consistency bound holds with a coefficient sufficiently small to guarantee:*

$$\mathcal{R}_{\Phi_{\log}}(h) \to 0 \implies \mathcal{R}(h) = 0.$$

*Proof.* Since $\mathcal{H}$ is strictly separable on the finite set $S$, there exists a base parameter $\theta_0$ and a minimum raw margin $\delta > 0$ such that the logit difference $z_{\theta_0}(x, y_{\text{win}}) - z_{\theta_0}(x, y_{\text{lose}}) \geq \delta$ for all tuples in $S$.

Consider the scaled logits $z_\alpha = \alpha z_{\theta_0}$ for $\alpha > 0$. As $\alpha \to \infty$, the policy $\pi_\alpha$ converges to a hard argmax distribution:

$$\lim_{\alpha \to \infty} \pi_\alpha(y_{\text{win}} \mid x) = 1, \quad \lim_{\alpha \to \infty} \pi_\alpha(y_{\text{lose}} \mid x) = 0.$$

The DPO implicit reward difference is given by:

$$\Delta h(x, y, y') = \beta \log \frac{\pi_\alpha(y_{\text{win}} \mid x)}{\pi_{\text{ref}}(y_{\text{win}} \mid x)} - \beta \log \frac{\pi_\alpha(y_{\text{lose}} \mid x)}{\pi_{\text{ref}}(y_{\text{lose}} \mid x)}.$$

The term $-\log \pi_\alpha(y_{\text{lose}} \mid x)$ dominates the expression, driving the magnitude of the score difference to infinity:

$$\lim_{\alpha \to \infty} w \cdot (h_\alpha(x, y) - h_\alpha(x, y')) = +\infty.$$

Consequently, for any finite dataset $S$, we can choose a scaling factor $\alpha$ large enough such that the effective margin $\gamma$ on all examples exceeds any arbitrary threshold. Applying the bound from Corollary 4.2, the consistency coefficient $\frac{1}{\beta\gamma}$ vanishes as $\gamma \to \infty$. Thus, on finite realizable domains, the "vacuous" bound identified in the negative result is overcome by the capacity of the model to drive margins to infinity. $\square$

## F. Proof of Proposition 5.3

**Proposition 5.3** (Vacuousness under Multiplicative Shift Invariance). *If a surrogate $\Phi$ satisfies $\Phi(u - \gamma) = C(\gamma)\Phi(u)$ for some $C(\gamma) > 0$, then for any $\gamma > 0$, the $\gamma$-shifted bound collapses to the unshifted bound ($\gamma = 0$). Specifically, the margin parameter cancels out, yielding: $\mathcal{R}(h) - \mathcal{R}^*(\mathcal{H}) + \mathcal{M}(\mathcal{H}) \leq \frac{1}{\Phi(0)}(\mathcal{R}_\Phi(h) - \mathcal{R}^*_\Phi(\mathcal{H}) + \mathcal{M}_\Phi(\mathcal{H})) + \mathcal{A}_0(\mathcal{H}).$*

*Proof.* Assume the invariance property holds: $\Phi(u - \gamma) = C(\gamma)\Phi(u)$. First, evaluating at $u = 0$ implies $\Phi(-\gamma) = C(\gamma)\Phi(0)$. Thus, the consistency coefficient becomes:

$$\frac{1}{\Phi(-\gamma)} = \frac{1}{C(\gamma)\Phi(0)}.$$

Second, the shifted surrogate error scales linearly:

$$\mathcal{R}_{\Phi_\gamma}(h) = \mathbb{E}[\Phi(w\Delta h - \gamma)] = \mathbb{E}[C(\gamma)\Phi(w\Delta h)] = C(\gamma)\mathcal{R}_\Phi(h).$$

This scaling applies strictly to the optimal error $\mathcal{R}^*_{\Phi_\gamma}(\mathcal{H}) = C(\gamma)\mathcal{R}^*_\Phi(\mathcal{H})$ and the minimizability gap $\mathcal{M}_{\Phi_\gamma}(\mathcal{H}) = C(\gamma)\mathcal{M}_\Phi(\mathcal{H})$. Consequently, the bracketed estimation term becomes:

$$\mathcal{R}_{\Phi_\gamma}(h) - \mathcal{R}^*_{\Phi_\gamma}(\mathcal{H}) + \mathcal{M}_{\Phi_\gamma}(\mathcal{H}) = C(\gamma)[\mathcal{R}_\Phi(h) - \mathcal{R}^*_\Phi(\mathcal{H}) + \mathcal{M}_\Phi(\mathcal{H})].$$

Third, the approximation gap $\mathcal{A}_\gamma(\mathcal{H})$ simplifies:

$$\mathcal{A}_\gamma(\mathcal{H}) = \frac{\mathbb{E}[\mathcal{C}^*_{\mathsf{L}_{\Phi_\gamma}}(\mathcal{H})]}{\Phi(-\gamma)} - \mathbb{E}[\mathcal{C}^*(\mathcal{H})] = \frac{C(\gamma)\,\mathbb{E}[\mathcal{C}^*_{\mathsf{L}_\Phi}(\mathcal{H})]}{C(\gamma)\Phi(0)} - \mathbb{E}[\mathcal{C}^*(\mathcal{H})] = \mathcal{A}_0(\mathcal{H}).$$

Substituting these components back into the $\gamma$-shifted bound (Theorem 5.2), the factor $C(\gamma)$ cancels in the first term, yielding the stated unshifted bound. $\qquad\square$

# G. Proof of Proposition 5.4

**Proposition 5.4** (Tightness of the $\gamma$-Shifted $\mathcal{H}$-Consistency Constant)**.** *Let $\Phi$ be non-increasing with $\lim_{u\to\infty}\Phi(u) = 0$. The coefficient $C = \frac{1}{\Phi(-\gamma)}$ is optimal: for any $C' < C$, there exist a distribution $\mathcal{D}$ and hypothesis set $\mathcal{H}$ such that the consistency bound fails for some $h \in \mathcal{H}$: $\mathcal{R}(h) - \mathcal{R}^*(\mathcal{H}) + \mathcal{M}(\mathcal{H}) > C'\Big[\mathcal{R}_{\Phi_\gamma}(h) - \mathcal{R}^*_{\Phi_\gamma}(\mathcal{H}) + \mathcal{M}_{\Phi_\gamma}(\mathcal{H})\Big] + \mathcal{A}_\gamma(\mathcal{H}).$*

*Proof.* Consider a distribution $\mathcal{D}$ supported on a single tuple $(x, y, y', w)$ with $w = 1$. The Bayes ranking error is 0. Let $\mathcal{H}$ be a hypothesis set containing a "bad" hypothesis $h_{\mathrm{bad}}$ with score difference $h_{\mathrm{bad}}(x, y) - h_{\mathrm{bad}}(x, y') = -\gamma$, and a sequence of "good" hypotheses $h_t$ with score difference $h_t(x, y) - h_t(x, y') = t > 0$.

Let $h_{\mathrm{boundary}}$ have score difference $h_{\mathrm{boundary}}(x, y) - h_{\mathrm{boundary}}(x, y') = 0$. The shifted score difference is $-\gamma$. Then:

$$\mathcal{R}(h_{\mathrm{boundary}}) = \Phi(0) = 1, \quad \mathcal{R}_{\Phi_\gamma}(h_{\mathrm{boundary}}) = \Phi(0 - \gamma) = \Phi(-\gamma).$$

Let the class $\mathcal{H}$ also contain a sequence of functions $h_t$ with score differences $h_t(x, y) - h_t(x, y') = t \to +\infty$. Then:

$$\mathcal{R}(h_t) = \Phi(t), \quad \mathcal{R}_{\Phi_\gamma}(h_t) = \Phi(t - \gamma).$$

As $t \to +\infty$, we have $\mathcal{R}_\Phi(h_t) \to 0$ and $\mathcal{R}_{\Phi_\gamma}(h_t) \to 0$. Thus: $\mathcal{R}^*(\mathcal{H}) = 0$ and $\mathcal{R}^*_{\Phi_\gamma}(\mathcal{H}) = 0$. Since $\mathcal{R}^*_{\Phi_\gamma}(\mathcal{H}) = \mathcal{R}^*(\mathcal{H}) = 0$, we have $\mathbb{E}\Big[\mathcal{C}^*_{\mathsf{L}_{\Phi_\gamma}}(\mathcal{H})\Big] = \mathbb{E}[\mathcal{C}^*(\mathcal{H})] = 0$, the approximation gap vanishes:

$$\mathcal{A}_\gamma(\mathcal{H}) = \frac{0}{\Phi(-\gamma)} - 0 = 0.$$

Also, the minimizability gaps vanish: $\mathcal{M}(\mathcal{H}) = \mathcal{M}_{\Phi_\gamma}(\mathcal{H}) = 0$. Substituting these values into the inequality with a hypothetical constant $C'$:

$$\mathcal{R}(h_{\mathrm{boundary}}) \le C'\mathcal{R}_{\Phi_\gamma}(h_{\mathrm{boundary}}) + 0 \Leftrightarrow 1 \le C'\Phi(-\gamma) \Leftrightarrow C' \ge \frac{1}{\Phi(-\gamma)}.$$

Thus, no constant smaller than $\frac{1}{\Phi(-\gamma)}$ can satisfy the bound universally. $\qquad\square$

# H. Proof of Theorem 5.6

**Theorem 5.6** (Structure-Aware $\gamma$-Shifted $\mathcal{H}$-Consistency Bound). *Let $\Phi: \mathbb{R} \to [0, \infty)$ be non-increasing with $\Phi(0) > 0$, and let $\Gamma(y, y') > 0$. Define the* inverse-margin weighted loss $\widetilde{\mathsf{L}}_{\Phi,\Gamma} = \frac{\mathsf{L}_{\Phi,\Gamma}}{\Phi(-\Gamma)}$. *Then, for any $h \in \mathcal{H}$:*

$$
\mathcal{R}(h) - \mathcal{R}^*(\mathcal{H}) + \mathcal{M}(\mathcal{H}) \le \left[ \mathcal{R}_{\widetilde{\mathsf{L}}_{\Phi,\Gamma}}(h) - \mathcal{R}^*_{\widetilde{\mathsf{L}}_{\Phi,\Gamma}}(\mathcal{H}) + \mathcal{M}_{\widetilde{\mathsf{L}}_{\Phi,\Gamma}}(\mathcal{H}) \right] + \mathcal{A}_\Gamma(\mathcal{H}),
$$

*where $\mathcal{A}_\Gamma(\mathcal{H}) = \mathbb{E}\left[ \mathcal{C}^*_{\widetilde{\mathsf{L}}_{\Phi,\Gamma}}(\mathcal{H}) \right] - \mathbb{E}\left[ \mathcal{C}^*(\mathcal{H}) \right]$ is the structure-aware approximation gap.*

*Proof.* Fix $h \in \mathcal{H}$. We begin with the pointwise analysis. Let $u = w \cdot (h(x, y) - h(x, y'))$ be the signed score difference. The 0-1 pairwise ranking loss is upper bounded by $1_{u \le 0}$. For a specific pair $(y, y')$, let $\gamma_{\text{loc}} = \Gamma(y, y')$. Since $\Phi$ is non-increasing and $\gamma_{\text{loc}} > 0$:

$$
u \le 0 \implies u - \gamma_{\text{loc}} \le -\gamma_{\text{loc}} \implies \Phi(u - \gamma_{\text{loc}}) \ge \Phi(-\gamma_{\text{loc}}) \ge \Phi(0) > 0.
$$

Dividing by the positive term $\Phi(-\gamma_{\text{loc}})$ and using the definition of the inverse-margin weighted loss $\widetilde{\mathsf{L}}_{\Phi,\Gamma}$, we obtain the pointwise bound:

$$
\mathsf{L}_{0-1}(h, x, y, y', w) \le 1_{u \le 0} \le \frac{\Phi(u - \gamma_{\text{loc}})}{\Phi(-\gamma_{\text{loc}})} = \widetilde{\mathsf{L}}_{\Phi,\Gamma}(h, x, y, y', w).
$$

Taking expectations over $w$ yields the inequality between conditional errors:

$$
\mathcal{C}(h) \le \mathcal{C}_{\widetilde{\mathsf{L}}_{\Phi,\Gamma}}(h).
$$

Since this holds for any $h \in \mathcal{H}$, it also holds for the infimum:

$$
\mathcal{C}^*(\mathcal{H}) = \inf_{h \in \mathcal{H}} \mathcal{C}(h) \le \inf_{h \in \mathcal{H}} \mathcal{C}_{\widetilde{\mathsf{L}}_{\Phi,\Gamma}}(h) = \mathcal{C}^*_{\widetilde{\mathsf{L}}_{\Phi,\Gamma}}(\mathcal{H}).
$$

This implies the approximation gap is non-negative: $\mathcal{A}_\Gamma(\mathcal{H}) \ge 0$. We now decompose the target conditional regret. For any $x, y, y'$:

$$
\begin{aligned}
\Delta \mathcal{C}_\mathcal{H}(h) &= \mathcal{C}(h) - \mathcal{C}^*(\mathcal{H}) \\
&\le \mathcal{C}_{\widetilde{\mathsf{L}}_{\Phi,\Gamma}}(h) - \mathcal{C}^*(\mathcal{H}) \\
&= \left[ \mathcal{C}_{\widetilde{\mathsf{L}}_{\Phi,\Gamma}}(h) - \mathcal{C}^*_{\widetilde{\mathsf{L}}_{\Phi,\Gamma}}(\mathcal{H}) \right] + \left[ \mathcal{C}^*_{\widetilde{\mathsf{L}}_{\Phi,\Gamma}}(\mathcal{H}) - \mathcal{C}^*(\mathcal{H}) \right] \\
&= \Delta \mathcal{C}_{\widetilde{\mathsf{L}}_{\Phi,\Gamma}, \mathcal{H}}(h) + \left[ \mathcal{C}^*_{\widetilde{\mathsf{L}}_{\Phi,\Gamma}}(\mathcal{H}) - \mathcal{C}^*(\mathcal{H}) \right].
\end{aligned}
$$

Taking the expectation over $(x, y, y')$:

- The LHS becomes $\mathbb{E}[\Delta \mathcal{C}_\mathcal{H}(h)] = \mathcal{R}(h) - \mathcal{R}^*(\mathcal{H}) + \mathcal{M}(\mathcal{H})$.

- The first term on the RHS becomes $\mathbb{E}[\Delta \mathcal{C}_{\widetilde{\mathsf{L}}_{\Phi,\Gamma}, \mathcal{H}}(h)] = \mathcal{R}_{\widetilde{\mathsf{L}}_{\Phi,\Gamma}}(h) - \mathcal{R}^*_{\widetilde{\mathsf{L}}_{\Phi,\Gamma}}(\mathcal{H}) + \mathcal{M}_{\widetilde{\mathsf{L}}_{\Phi,\Gamma}}(\mathcal{H})$.

- The second term on the RHS becomes the structure-aware approximation gap $\mathcal{A}_\Gamma(\mathcal{H})$.

Combining these yields the stated bound. $\qquad\square$

# I. Proof of Proposition 6.1 and Proposition 6.2

**Proposition 6.1** (Vanishing Gap under Infinite Capacity). *Let $\Phi: \mathbb{R} \to [0, \infty)$ satisfy $\Phi(-\gamma) > 0$ and $\lim_{u \to +\infty} \Phi(u) = 0$. Let $\mathcal{H}$ be a hypothesis set closed under positive scalar multiplication (i.e., $h \in \mathcal{H} \implies \alpha h \in \mathcal{H}$ for all $\alpha > 0$). If $\mathcal{H}$ is capable of perfect ranking on the support of $\mathcal{D}$ (i.e., $\mathcal{R}^*(\mathcal{H}) = 0$), then for any finite margin $\gamma > 0$:*

$$
\lim_{\alpha \to \infty} \mathcal{A}_\gamma(\mathcal{H}_\alpha) = 0, \tag{3}
$$

*where $\mathcal{H}_\alpha$ denotes the scaled hypothesis set.*

*Proof.* Since $\mathcal{R}^*(\mathcal{H}) = 0$, there exists $h \in \mathcal{H}$ such that $w \cdot (h(x,y) - h(x,y')) > 0$ almost surely. Let $u(x) = w \cdot (h(x,y) - h(x,y'))$ denote this signed score difference. Consider the scaled hypothesis $h_\alpha = \alpha h$. The shifted surrogate error is:

$$\mathcal{R}_{\Phi_\gamma}(h_\alpha) = \mathop{\mathbb{E}}_{(x,y,y',w) \sim \mathcal{D}} [\Phi(\alpha u(x) - \gamma)].$$

Since $u(x) > 0$, as $\alpha \to +\infty$, the argument $\alpha u(x) - \gamma \to +\infty$. Since $\lim_{u \to +\infty} \Phi(u) = 0$, by the dominated convergence theorem, $\mathcal{R}_{\Phi_\gamma}(h_\alpha) \to 0$. Consequently, we have $\mathcal{R}^*_{\Phi_\gamma}(\mathcal{H}) = \mathcal{R}^*(\mathcal{H}) = 0$. This implies that the expected best-in-class conditional errors vanish, i.e., $\mathbb{E}\left[\mathcal{C}^*_{\mathsf{L}_{\Phi_\gamma}}(\mathcal{H})\right] = \mathbb{E}[\mathcal{C}^*(\mathcal{H})] = 0$, yielding: $\mathcal{A}_\gamma(\mathcal{H}) = \frac{0}{\Phi(-\gamma)} - 0 = 0$. $\square$

**Proposition 6.2** (Penalty under Bounded Capacity). *Conversely, if the hypothesis set $\mathcal{H}$ has bounded outputs (e.g., due to the activation function used or strict spectral normalization) such that $\sup_{(h,x,y,y') \in \mathcal{H} \times \mathcal{X} \times \mathcal{Y} \times \mathcal{Y}} |h(x,y) - h(x,y')| \leq K$, then for any margin $\gamma > K$:*

$$\mathcal{A}_\gamma(\mathcal{H}) \geq \frac{\Phi(K - \gamma)}{\Phi(-\gamma)} > 0, \tag{4}$$

*even if the true ranking error $\mathcal{R}^*(\mathcal{H})$ is zero.*

*Proof.* Assume $\mathcal{R}^*(\mathcal{H}) = 0$. The model correctly ranks all pairs, but the maximum score difference is $K$. The margin-shifted argument is at most $K - \gamma$. Since $\gamma > K$, this argument is negative. Since $\Phi$ is non-increasing, $\Phi(u - \gamma) \geq \Phi(K - \gamma) \geq \Phi(0) > 0$. Thus, the best-in-class surrogate conditional error is lower-bounded by $\Phi(K - \gamma)$. Substituting this into the definition of $\mathcal{A}_\gamma$ yields the strictly positive lower bound. $\square$

## J. Proof of Proposition 6.5

**Proposition 6.5** (Monotonicity of Profile with Tail Degree). *Let $\gamma > K > 0$. The Margin-Capacity Profile for the Polynomial Hinge Loss of degree $k$ is given by:*

$$\rho_k(\gamma, K) = \left(\frac{1 + \gamma - K}{1 + \gamma}\right)^k = \left(1 - \frac{K}{1 + \gamma}\right)^k. \tag{7}$$

*For Logistic loss (linear tail), the profile is approximately:*

$$\rho_{\Phi_{\log}}(\gamma, K) \approx 1 - \frac{K}{\gamma}. \tag{8}$$

*Proof.* For the Polynomial Hinge loss, $\Phi_{\text{poly-}k}(-\gamma) = (1 + \gamma)^k$ and $\Phi_{\text{poly-}k}(K - \gamma) = (1 - (K - \gamma))^k = (1 + \gamma - K)^k$. The ratio yields the result immediately. For the Logistic loss, $\Phi_{\log}(-\gamma) \approx \beta\gamma$ and $\Phi_{\log}(K - \gamma) \approx \beta(\gamma - K)$. The ratio is $\frac{\beta(\gamma - K)}{\beta\gamma} = 1 - \frac{K}{\gamma}$. $\square$

## K. Proof of Proposition 6.6

**Proposition 6.6** (Profile of Bounded Losses). *For any bounded surrogate loss where $\lim_{u \to -\infty} \Phi(u) = C > 0$, the Margin-Capacity Profile satisfies: $\lim_{\gamma \to \infty} \rho_\Phi(\gamma, K) = 1$.*

*Proof.* As $\gamma \to \infty$, both the numerator argument $K - \gamma$ and the denominator argument $-\gamma$ approach $-\infty$. Thus, both $\Phi(K - \gamma)$ and $\Phi(-\gamma)$ converge to the limit constant $C$. The ratio converges to 1. $\square$

## L. Proof of Proposition 6.7

**Proposition 6.7** (Optimal Margin for Bounded Models). *For $\Phi_{\log}(u) = \log(1 + e^{-\beta u})$ and a hypothesis set with score capacity $K$, the optimal margin $\gamma^*$ that minimizes the $\gamma$-shifted $\mathcal{H}$-consistency bound is the solution to:*

$$\gamma^* \approx K + \frac{1}{\beta} \log\left(\frac{\epsilon}{\beta K}\right). \tag{10}$$

*Proof.* For the logistic loss with parameter $\beta$, we use the approximations for large margins ($u \ll 0 \Rightarrow \Phi_{\log}(u) \approx -\beta u$, $u \gg 0 \Rightarrow \Phi_{\log}(u) \approx e^{-\beta u}$):

1. Coefficient: $\Phi_{\log}(-\gamma) = \log(1 + e^{\beta\gamma}) \approx \beta\gamma$.

2. Approximation Gap Numerator: $\Phi_{\log}(K - \gamma)$. Since we typically set $\gamma \approx K$ to push capacity, $K - \gamma$ is small. If we push $\gamma > K$, then $\Phi_{\log}(K - \gamma) \approx \beta(\gamma - K)$. If $\gamma < K$, $\Phi_{\log}(K - \gamma) \approx e^{-\beta(K-\gamma)}$.

Consider the regime where we push the margin close to capacity ($\gamma \approx K$). The bound simplifies to:

$$B(\gamma) \approx \frac{\epsilon}{\beta\gamma} + \frac{e^{-\beta(K-\gamma)}}{\beta\gamma}.$$

To find the minimum, take the derivative with respect to $\gamma$ and set to 0:

$$\frac{\partial B}{\partial \gamma} = -\frac{\epsilon}{\beta\gamma^2} - \frac{e^{-\beta(K-\gamma)}}{\beta\gamma^2} + \frac{e^{-\beta(K-\gamma)}}{\gamma} = 0.$$

Multiplying by $\beta\gamma^2$:

$$-\epsilon - e^{-\beta(K-\gamma)} + \beta\gamma e^{-\beta(K-\gamma)} = 0 \implies e^{-\beta(K-\gamma)}(\beta\gamma - 1) = \epsilon.$$

Assuming $\beta\gamma \gg 1$ (large margin/low temperature), we approximate $\beta\gamma - 1 \approx \beta\gamma \approx \beta K$.

$$e^{-\beta(K-\gamma)} \cdot \beta K \approx \epsilon \Rightarrow -\beta(K - \gamma) \approx \log\left(\frac{\epsilon}{\beta K}\right).$$

Solving for $\gamma$:

$$K - \gamma \approx -\frac{1}{\beta}\log\left(\frac{\epsilon}{\beta K}\right) \Rightarrow \gamma^* \approx K + \frac{1}{\beta}\log\left(\frac{\epsilon}{\beta K}\right).$$

$\square$

## M. Experimental Details

We provide the comprehensive configuration details for the experiments presented in Section 7.

**Controlled Validation Settings.** For the *Synonym Stress Test*, we fine-tuned Llama-3-8B using LoRA with rank $r = 16$, alpha $\alpha = 16$, and a learning rate of $5 \times 10^{-5}$. For SA-DPO, the structure-aware margin scaling factor was set to $\tau = 5.0$. For the *Margin-Capacity Analysis*, we used the Anthropic HH-RLHF dataset (Bai et al., 2022). We used a constrained LoRA configuration ($r = 8, \alpha = 16$) to limit model capacity. Models were trained for an extended duration of 550 steps with a learning rate of $2 \times 10^{-5}$ to ensure convergence behavior was strictly due to loss geometry rather than insufficient training.

**Real-World Evaluation Settings.** We fine-tuned Llama-3-8B-Instruct (4-bit quantized) on a 20k subset of the UltraFeedback dataset (Cui et al., 2024). We applied LoRA ($r = 32, \alpha = 32$) to all linear projection layers. Training was conducted for 1,200 steps with a batch size of 16 and a learning rate of $1 \times 10^{-4}$, using a cosine schedule with 5% warmup.

**Argilla DPO-Mix-7k Settings.** We fine-tuned Qwen2.5-7B-Instruct (Yang et al., 2024) using LoRA ($r = 32, \alpha = 32$) for 1,000 steps with a batch size of 16 and a learning rate of $5 \times 10^{-5}$. The margin configurations ($\gamma$ and $\tau$) were aligned using the same median distance approach as the UltraFeedback experiments. The downstream generation evaluation was conducted using 200 hold-out prompts from MT-Bench (Zheng et al., 2023), scored by the `PairRM` model (Jiang et al., 2023).

- **SimPO Instantiation:** We performed a grid search on a hold-out validation set to select the optimal fixed margin $\gamma = 0.7$, fixing $\beta = 1.0$.

- **SA-DPO Instantiation:** We defined the structure-aware margin $\Gamma(y, y') = \tau\Delta(y, y')$, using the cosine distance $\Delta$ from `BGE-Large-v1.5` embeddings (Xiao et al., 2023). We calibrated $\tau \approx 3.4$ to align the average margin on all training pairs with SimPO ($\mathbb{E}[\Gamma] = \tau\mathbb{E}[\Delta] \approx 0.7$), using the same $\beta = 1.0$.

*Table 4.* Correspondence between divergences and surrogate ranking losses.

| Regularizer ($\psi$) | Divergence ($B_\Psi$) | Surrogate $\Phi(u)$ | Method |
|---|---|---|---|
| Shannon Entropy | Kullback-Leibler | $\log(1 + e^{-\beta u})$ | DPO (Rafailov et al., 2023) |
| Squared $L_2$ Norm | Squared Euclidean | $\left[u - \frac{\beta}{2}\right]^2$ | IPO (Azar et al., 2024) |
| Tsallis Entropy ($q$-Log) | Tsallis Divergence | $\Phi_q(u)$ (Sparsemax) | Sparse-DPO (Martins & Astudillo, 2016) |

## N. Theoretical Extensions and Discussion

In this section, we discuss the implications of our framework regarding regularization, convex optimization, and standard assumptions in preference learning.

### N.1. Generalization to Bregman Divergences

Our analysis of ranking consistency is not limited to the KL-divergence formulation of DPO. It extends to the broader class of $\Psi$-regularized preference optimization methods.

Consider the general regularized objective with a Bregman divergence $B_\Psi$ generated by a strictly convex function $\psi$:

$$\max_\pi \mathbb{E}_{(x,y)\sim\pi}\left[r(x,y)\right] - \beta \mathbb{E}_x[B_\Psi(\pi(\cdot|x), \pi_{\text{ref}}(\cdot|x))]. \tag{11}$$

The optimal policy for such problems satisfies a specific link function relating rewards to probability ratios. When converted into a preference ranking loss, this induces a specific surrogate function $\Phi_\psi$.

**Universality of the Margin Shift.** Theorem 5.2 relies only on the convexity and monotonicity of the surrogate $\Phi$. Therefore, the proposed fix is universal. For any divergence $B_\Psi$ inducing a convex surrogate $\Phi_\psi$, the consistent margin-shifted objective is:

$$L_{\text{margin}-\psi}(h) = \Phi_\psi(w \cdot \Delta h - \gamma). \tag{12}$$

Notably, the Identity Preference Optimization (IPO) method (Azar et al., 2024) minimizes a squared loss equivalent to $\Phi_{\text{sq}}(u) = (u - \mu)^2$. In our framework, this can be viewed as a margin-shifted surrogate where the target margin is intrinsic to the loss definition. This explains the empirical robustness of IPO: it inherently enforces the separation gap $\gamma$ required for $\mathcal{H}$-consistency.

#### N.1.1. Equivalence

To formalize the connection between the reward maximization objective and our ranking surrogate, we first define the *score function* induced by the regularizer.

**Definition N.1** (Bregman Score Function). Let $\psi$ be a strictly convex function generating the Bregman divergence $B_\Psi$. We define the *score function* $h_\Psi(\pi(\cdot|x), y)$ as the $y$-th component of the gradient difference relative to the reference policy:

$$h_\Psi(\pi(\cdot|x), y) \triangleq \nabla\psi(\pi(\cdot|x))_y - \nabla\psi(\pi_{\text{ref}}(\cdot|x))_y. \tag{13}$$

For separable regularizers (e.g., entropy), this simplifies to element-wise operations on $\pi(y|x)$.

We also formalize the probabilistic assumption linking arbitrary loss functions to likelihood maximization.

**Definition N.2** (Surrogate-Induced Preference Model). Let $\Phi : \mathbb{R} \to \mathbb{R}$ be a convex surrogate loss function. We define the *Surrogate-Induced Preference* model induced by $\Phi$ as:

$$P(y_w \succ y_l \mid x) = \sigma_\Phi(r(x, y_w) - r(x, y_l)), \tag{14}$$

where the link function $\sigma_\Phi$ is defined by the inverse negative log-likelihood:

$$\sigma_\Phi(u) = \exp(-\Phi(u)). \tag{15}$$

We now show that minimizing the generic ranking surrogate is equivalent to maximizing the regularized reward under this probabilistic model. See Appendix O for a proof.

**Theorem N.3** (Equivalence of Regularized RLHF and $\Phi$-Ranking). *Let $\pi^*$ be the optimal policy of the $\Psi$-regularized RLHF objective:* $\max_\pi \mathcal{L}_{\text{RLHF}}(\pi) =$

$$\mathbb{E}_{x \sim \mathcal{D}}\left[\mathbb{E}_{y \sim \pi(\cdot|x)}[r(x,y)] - \beta \mathsf{B}_\Psi(\pi(\cdot|x)\|\pi_{\text{ref}}(\cdot|x))\right]. \tag{16}$$

*Assume the ground-truth preferences follow the Surrogate-Induced Preference model (Definition N.2) induced by a surrogate $\Phi$. Then, minimizing the ranking objective:* $\mathcal{L}_{\text{Rank}}(\pi) =$

$$\mathbb{E}_{(x,y_w,y_l) \sim \mathcal{D}_{\text{pref}}}\left[\Phi(\beta\left(h\Psi(\pi(\cdot|x), y_w) - h_\Psi(\pi(\cdot|x), y_l)\right))\right]$$

*yields the same optimal solution $\pi^*$.*

**Remark on IPO.** It is important to note that while Identity Preference Optimization (IPO) (Azar et al., 2024) minimizes a squared loss equivalent to $\Phi(u) = (u - \beta/2)^2$, it retains the entropic regularization geometry where $\psi(\pi)$ is the Shannon entropy. This implies $h_\Psi$ remains logarithmic ($h_\Psi = \log \pi - \log \pi_{\text{ref}}$), distinct from methods using Euclidean regularization (where $h_\Psi$ would be linear).

## N.2. Regularization and Optimization Landscape

We highlight the difficulty of enforcing consistency through regularization on the scoring function itself versus the model parameters.

**Impossibility of Convex Score Regularization.** One might attempt to enforce separation by adding a regularization term $R(h)$ to the loss that penalizes small score differences. However, to force scores away from zero (i.e., towards $+\infty$ or $-\infty$), the penalty function would need to be minimized at infinity and maximized at zero (e.g., an inverted Gaussian). Such a function is inherently non-convex. Thus, no convex regularization on the score output space can enforce a margin.

**Implicit Bias of Parameter Regularization.** Conversely, $L_2$ regularization on the model parameters $\theta$ can aid consistency through implicit bias. For the score differences to be close to zero across the board (inconsistency), the parameter norm $\|\theta\|$ typically needs to vanish. By regularizing parameters or training with gradient descent on exponential losses, the parameters are driven to increase in the direction of the max-margin solution (Soudry et al., 2018), implicitly maximizing the separation.

**Slack Variable Interpretation.** Using the $\gamma$-shifted loss introduced in Section 5 is equivalent to imposing a hard margin constraint with slack variables. Specifically, minimizing $\sum \Phi(w_i \Delta h_i - \gamma)$ is equivalent to the following soft-margin optimization problem:

$$\min_{h \in \mathcal{H}, \xi} \sum \Phi(\xi_i) \text{ subject to } w_i(h(x_i, y_i) - h(x_i, y'_i)) \geq \gamma - \xi_i.$$

This view reconciles the need for a margin $\gamma$ (for consistency) with the need for a convex relaxation (for tractability).

## N.3. Relation to the Bradley-Terry Assumption

Many works assume the preference data follows a Bradley-Terry (BT) model, where $\eta(x,y,y') = \sigma(r^*(x,y) - r^*(x,y'))$. In our framework, the BT assumption, combined with the assumption that $r^* \in \mathcal{H}$, is equivalent to assuming the surrogate minimizability gap vanishes ($\mathcal{M}_{\Phi_{\log}}(\mathcal{H}) = 0$) (See Appendix P for a proof). Our analysis is more general, as it provides consistency guarantees even in the agnostic setting where the BT model is misspecified ($\mathcal{M}_{\Phi_{\log}}(\mathcal{H}) > 0$), accounting for the approximation error via the gap terms.

**Proposition N.4** (Bradley-Terry Equivalence). *Assume the preference distribution $\mathcal{D}$ follows the Bradley-Terry (BT) model, such that the true conditional preference probability is given by $\eta(x,y,y') = \sigma(r^*(x,y) - r^*(x,y'))$ for some latent reward function $r^*$, where $\sigma$ is the sigmoid function. If the hypothesis set is realizable (i.e., $r^* \in \mathcal{H}$), then the minimizability gap for the logistic surrogate loss $\Phi_{\log}$ vanishes:* $\mathcal{M}_{\Phi_{\log}}(\mathcal{H}) = 0$.

**Remark on Non-Transitivity.** We note that our analysis, like the DPO framework itself, assumes the existence of a global scoring function $h$. In scenarios where human preferences are inherently non-transitive (cyclic), no scalar reward model can perfectly capture the underlying relation. While alternative approaches based on pairwise classification and randomized sorting algorithms (Ailon & Mohri, 2010) exist to handle such cycles, they effectively abandon the reward modeling paradigm. Our work focuses on ensuring consistency within the scoring paradigm that dominates current LLM alignment.

# O. Proof of Theorem N.3

**Theorem N.3** (Equivalence of Regularized RLHF and $\Phi$-Ranking). *Let $\pi^*$ be the optimal policy of the $\Psi$-regularized RLHF objective:* $\max_\pi \mathcal{L}_{\mathrm{RLHF}}(\pi) =$

$$\mathbb{E}_{x \sim \mathcal{D}}\left[\mathbb{E}_{y \sim \pi(\cdot|x)}[r(x,y)] - \beta \mathsf{B}_\Psi(\pi(\cdot|x)\|\pi_{\mathrm{ref}}(\cdot|x))\right]. \tag{16}$$

*Assume the ground-truth preferences follow the Surrogate-Induced Preference model (Definition N.2) induced by a surrogate $\Phi$. Then, minimizing the ranking objective:* $\mathcal{L}_{\mathrm{Rank}}(\pi) =$

$$\mathbb{E}_{(x,y_w,y_l) \sim \mathcal{D}_{\mathrm{pref}}}\left[\Phi\big(\beta\left(h\Psi(\pi(\cdot|x),y_w) - h_\Psi(\pi(\cdot|x),y_l)\right)\big)\right]$$

*yields the same optimal solution $\pi^*$.*

*Proof.* We first derive the analytical solution for the RLHF objective (Eq. 16). Since the expectation over $x$ is linear, we can solve the optimization problem pointwise for each context $x$. We introduce a Lagrange multiplier $\lambda(x)$ to satisfy the simplex constraint $\sum_y \pi(y|x) = 1$. The Lagrangian is:

$$\mathcal{L}(\pi,\lambda) = \sum_y \pi(y|x)r(x,y) - \beta\mathsf{B}_\Psi(\pi(\cdot|x)\|\pi_{\mathrm{ref}}(\cdot|x)) + \lambda(x)\left(1 - \sum_y \pi(y|x)\right). \tag{17}$$

Taking the gradient with respect to a specific probability mass $\pi(y|x)$ and setting it to zero (KKT conditions):

$$r(x,y) - \beta(\nabla\psi(\pi(\cdot|x))_y - \nabla\psi(\pi_{\mathrm{ref}}(\cdot|x))_y) - \lambda(x) = 0. \tag{18}$$

Rearranging terms and using the definition of the score function $h_\Psi$, we obtain the relationship between the latent reward and the optimal policy $\pi^*$:

$$r(x,y) = \beta h_\Psi(\pi^*(\cdot|x),y) + \lambda(x). \tag{19}$$

Here, $\lambda(x)$ serves as the context-dependent partition function (normalization constant). Next, we consider the preference modeling task. Under the Surrogate-Induced Preference assumption (Definition N.2), maximizing the likelihood of the preference data $\mathcal{D}_{\mathrm{pref}}$ is equivalent to minimizing the negative log-likelihood, which by definition matches the expectation of the surrogate $\Phi$:

$$\min_\pi \mathbb{E}_{(x,y_w,y_l) \sim \mathcal{D}_{\mathrm{pref}}}\left[-\log P(y_w \succ y_l \mid x)\right] \equiv \min_\pi \mathbb{E}_{(x,y_w,y_l) \sim \mathcal{D}_{\mathrm{pref}}}\left[\Phi(r(x,y_w) - r(x,y_l))\right]. \tag{20}$$

We substitute the optimality condition for the reward $r(x,y)$ from Eq. (19) into the margin term inside $\Phi$. For any context $x$:

$$r(x,y_w) - r(x,y_l) = (\beta h_\Psi(\pi^*(\cdot|x),y_w) + \lambda(x)) - (\beta h_\Psi(\pi^*(\cdot|x),y_l) + \lambda(x)) \tag{21}$$

$$= \beta(h_\Psi(\pi^*(\cdot|x),y_w) - h_\Psi(\pi^*(\cdot|x),y_l)). \tag{22}$$

Crucially, the context-dependent normalization term $\lambda(x)$ cancels out in the difference. Therefore, minimizing the loss defined solely on the policy scores:

$$\mathcal{L}_{\mathrm{Rank}}(\pi) = \mathbb{E}_{(x,y_w,y_l) \sim \mathcal{D}_{\mathrm{pref}}}\left[\Phi(\beta\Delta h_\Psi(\pi(\cdot|x)))\right] \tag{23}$$

is mathematically equivalent to optimizing the likelihood of the reward model consistent with the original RLHF problem. $\square$

# P. Proof of Proposition N.4

**Proposition N.4** (Bradley-Terry Equivalence). *Assume the preference distribution $\mathcal{D}$ follows the Bradley-Terry (BT) model, such that the true conditional preference probability is given by $\eta(x,y,y') = \sigma(r^*(x,y) - r^*(x,y'))$ for some latent reward function $r^*$, where $\sigma$ is the sigmoid function. If the hypothesis set is realizable (i.e., $r^* \in \mathcal{H}$), then the minimizability gap for the logistic surrogate loss $\Phi_{\log}$ vanishes: $\mathcal{M}_{\Phi_{\log}}(\mathcal{H}) = 0$.*

*Proof.* The minimizability gap is defined as $\mathcal{M}_\Phi(\mathcal{H}) = \mathcal{R}_\Phi^*(\mathcal{H}) - \mathcal{R}_\Phi^*(\mathcal{H}_{\mathrm{all}})$, where $\mathcal{H}_{\mathrm{all}}$ is the set of all measurable functions. For the logistic loss $\mathsf{L}_{\Phi_{\log}}(h) = -\log \sigma(w\Delta h)$, it is a standard result that the pointwise minimizer $h^*$ over the space of all measurable functions satisfies the log-odds relationship:

$$h^*(x,y) - h^*(x,y') = \log \frac{\eta(x,y,y')}{1 - \eta(x,y,y')}.$$

Under the Bradley-Terry assumption, substituting $\eta = \sigma(\Delta r^*)$, the log-odds term simplifies exactly to the reward difference:

$$\log \frac{\sigma(\Delta r^*)}{\sigma(-\Delta r^*)} = r^*(x,y) - r^*(x,y').$$

Thus, the Bayes optimal scoring function is simply the true reward function $r^*$. Since we assume realizability ($r^* \in \mathcal{H}$), the hypothesis set $\mathcal{H}$ contains the Bayes optimal minimizer. Therefore, $\inf_{h\in\mathcal{H}} \mathcal{R}_{\Phi_{\log}}(h) = \inf_{h\in\mathcal{H}_{\mathrm{all}}} \mathcal{R}_{\Phi_{\log}}(h)$, implying that the gap $\mathcal{M}_{\Phi_{\log}}(\mathcal{H})$ is zero. $\qquad\square$

