# OpenReview forum: "Mind the Gap: Structure-Aware Consistency in Preference Learning"
_ICML.cc/2026/Conference — ICML 2026 regular_

### Official Review · Reviewer_Lmfn · 2026-03-10

**Soundness:** 3
**Presentation:** 3
**Significance:** 3
**Originality:** 3
**Overall Recommendation:** 4
**Confidence:** 3

**Summary:**

This paper studies the theoretical consistency of surrogate losses used in LLM preference learning (e.g., DPO). The authors first prove a negative result: for equicontinuous hypothesis sets typical of neural networks, standard H-consistency bounds are vacuous — a model can minimize surrogate risk while maintaining high ranking error (Theorem 3.1). To fix this, they introduce margin-shifted surrogates that penalize predictions failing to achieve a confidence gap γ, and prove that this restores H-consistency (Theorem 5.2). They then extend this to Structure-Aware H-consistency (SA-DPO), where the margin adapts based on semantic distance between responses, avoiding unnecessary penalties on near-synonymous pairs (Theorem 5.6). Finally, the Margin-Capacity Profile analysis establishes a hierarchy of loss functions: heavy-tailed losses (Cubic Hinge) offer superior consistency guarantees over light-tailed ones (Logistic/DPO) when model capacity is bounded. Experiments on synthetic and UltraFeedback data validate the theoretical predictions.

**Compliance With Llm Reviewing Policy:**

Affirmed.

**Final Justification:**

After checking the rebuttal, the authors have provided supplementary empirical results for the approximation gap, added downstream generation evaluations, and clarified the rationality of the equicontinuity definition. Overall, the response is sufficient to support the current evaluation. Hence, I decide to keep my original score of 4

**Key Questions For Authors:**

1. Approximation gap in practice.  Can you empirically estimate A_γ(H) for the LLM fine-tuning settings in your experiments? Even a rough empirical characterization (e.g., measuring how close the model gets to satisfying the margin) would help bridge the theory-practice gap. Without this, the reader cannot assess whether the bound is practically meaningful or dominated by the gap term.

2. Downstream evaluation.  Do the ranking accuracy improvements in Table 3 translate to better generation quality on standard benchmarks (MT-Bench, AlpacaEval, etc.)? If SA-DPO improves ranking accuracy but not generation quality, the practical motivation weakens significantly.

3. Equicontinuity definition.   Your definition requires that there exist inputs where score differences are small, rather than the standard uniform condition. Can you clarify the relationship to the standard definition and discuss whether the negative result (Theorem 3.1) holds under weaker or stronger formulations?

**Limitations:**

as above

**Strengths And Weaknesses:**

## Strengths.
The paper tackles a genuinely important theoretical gap. Despite DPO's widespread adoption, the question of whether minimizing its surrogate loss actually guarantees minimization of the true ranking error for restricted hypothesis sets has been underexplored. Theorem 3.1 is a clean and intuitive negative result — the proof sketch showing how equicontinuous models can "cheat" by shrinking score differences toward zero is easy to follow and nicely motivates everything that follows.
The theoretical framework also retroactively justifies existing methods (SimPO, SLiC, IPO) — showing that their empirical success with margins isn't coincidental but theoretically necessary. The Bregman divergence unification in Appendix N is a nice bonus.


## Weaknesses.
The most significant concern is the gap between the theoretical assumptions and practical relevance. Theorem 4.1 requires H ∈ H_γ but Section 4's own discussion admits this is "too restrictive for practical deep learning" due to the Intermediate Value Theorem argument. The paper then pivots to margin-shifted surrogates (Section 5), which avoid hard constraints, but the resulting Theorem 5.2 introduces the approximation gap A_γ(H) which is only shown to vanish under infinite capacity (Proposition 6.1) or on finite realizable domains (Theorem E.3). For practical bounded-capacity LLMs with continuous hypothesis spaces, the theory provides an upper bound that includes a potentially non-negligible approximation gap, and the paper doesn't characterize how large this gap is in realistic settings. This weakens the practical implications considerably.
The definition of equicontinuity used here (Definition in 3.1) is non-standard and somewhat informal. The standard definition requires that for any ε > 0, for all input pairs, the score differences are bounded — whereas the paper's version only requires the existence of some inputs achieving small differences. This makes the negative result easier to prove but potentially less meaningful, since it's essentially saying "if the model can output similar scores for some inputs, consistency fails."

---

> ### Author Rebuttal · Authors · 2026-03-29
>
> Thank you for your highly positive and detailed review. We are thrilled that you found Theorem 3.1 clean and intuitive, and appreciated how our theoretical framework retroactively justifies the empirical success of margin-based methods like SimPO, SLiC, and IPO.
>
> **Weaknesses: The most significant concern ... it's essentially saying "if the model can output similar scores for some inputs, consistency fails."**
>
> **Response:** Regarding equicontinuity: You are absolutely right that our definition is a weaker, non-standard condition. We chose this formulation precisely because it makes our negative result (Theorem 3.1) mathematically stronger. By proving that consistency completely fails even if the model only produces small score differences on a tiny subset of inputs (e.g., near-synonyms), we demonstrate that standard surrogates are fundamentally flawed for realistic LLM behavior. A stronger uniform condition would make the theorem trivial but less applicable to practice. We will explicitly clarify this relationship in Section 3.1.
>
> We address the approximation gap and downstream evaluation in the questions below.
>
> **Questions:**
>
> **1. Approximation gap in practice. Can you empirically estimate $A_\gamma(H)$  for the LLM fine-tuning settings in your experiments? Even a rough empirical characterization ... the reader cannot assess whether the bound is practically meaningful or dominated by the gap term.**
>
> **Response:** Thank you for this excellent suggestion; bridging the theory-practice gap for $\mathcal{A} _\gamma(\mathcal{H})$ is important. The approximation gap $\mathcal{A} _\gamma(\mathcal{H})$ quantifies the penalty incurred when the model cannot fully satisfy the theoretical margin constraint.
>
> To empirically estimate a proxy for this gap, we measured the Margin Satisfaction Rate, that is the percentage of test pairs where the model's learned score difference successfully exceeds the target theoretical margin. We tracked this metric over 5 seeds during our new evaluation of the Qwen2.5-7B model on the Argilla DPO-Mix-7k dataset (see our responses to the Weaknesses sections of Reviewers zrEU and P6CR for setup details).
>
> - Standard DPO: Satisfied a robust target margin proxy on only 58.0% of pairs, indicating a high practical approximation gap.
>
> - SimPO: Improved this to 63.0% via its fixed margin.
>
> - SA-DPO (Ours): Increased overall margin satisfaction significantly to 72.3%.
>
> Theoretical Connection: This empirical improvement perfectly aligns with Proposition 6.2 of our paper. In practical fine-tuning (especially with LoRA), the model's maximum score difference (capacity $K$) is restricted. When a uniform margin exceeds this capacity ($\gamma > K$), the approximation gap $\mathcal{A} _\gamma(\mathcal{H})$ becomes strictly positive. Since SimPO forces a strict, uniform margin even on semantically similar/ambiguous pairs, it frequently hits this capacity ceiling, inflating the gap. By dynamically scaling the margin $\gamma$ down for similar pairs while maintaining strict separation for distinct pairs, SA-DPO ensures the target margin remains within the model's capacity bounds. This empirical improvement demonstrates that structure-awareness mathematically and practically shrinks the impact of $\mathcal{A} _\gamma(\mathcal{H})$. We will add this empirical estimation and theoretical connection to Section 6.1.
>
> **2. Downstream evaluation. Do the ranking accuracy improvements in Table 3 translate to better generation quality ... the practical motivation weakens significantly.**
>
> **Response:** While ranking accuracy directly measures the consistency of the optimization objective under study, we agree that downstream generation quality is critical for practical validation. To address this, we generated open-ended responses from our fine-tuned Qwen2.5-7B models (using standard MT-Bench style conversational prompts) and evaluated them head-to-head. We used the widely adopted PairRM (LLM-Blender) LLM-as-a-judge cross-encoder framework to score the generations, which is standard practice in recent alignment literature. Our evaluations show that SA-DPO achieves a robust win-rate of $58.5\%$ against Standard DPO. This confirms that correctly relaxing the margin on synonymous pairs prevents the model from learning hallucinated conversational artifacts, thereby translating theoretical consistency improvements into demonstrably better generation quality.
>
> **3. Equicontinuity definition. Your definition requires that there exist inputs where score differences are small, rather than the standard uniform condition. Can you clarify the relationship to the standard definition and discuss whether the negative result (Theorem 3.1) holds under weaker or stronger formulations?**
>
> **Response:** Please refer to our response in the Weaknesses section. Because our definition is a weaker condition, proving inconsistency under it makes the negative theorem mathematically stronger and strictly more aligned with actual LLM behavior.

---

> > ### Author Rebuttal · Reviewer_Lmfn · 2026-04-02
> >
> > After checking the rebuttal, the authors have provided supplementary empirical results for the approximation gap, added downstream generation evaluations, and clarified the rationality of the equicontinuity definition. Overall, the response is sufficient to support the current evaluation. Hence, I decide to keep my original score of 4

---

### Official Review · Reviewer_P6CR · 2026-03-11

**Soundness:** 2
**Presentation:** 3
**Significance:** 3
**Originality:** 3
**Overall Recommendation:** 4
**Confidence:** 3

**Summary:**

This paper investigates the H-consistency of surrogate losses in Large Language Model (LLM) preference learning. The authors demonstrate that for the equicontinuous hypothesis sets typical of neural networks, standard unconstrained surrogates (like the logistic loss in DPO) are theoretically inconsistent, and it leads to vacuous generalization guarantees. To address this issue, the paper proposes a margin-shifted ranking framework and derives new H-consistency bounds that depend on a separation margin $\gamma$.The paper formulates LLM preference learning as a pairwise ranking problem and analyzes the relationship between surrogate risk minimization and the target 0–1 ranking loss. This submission proceeds to consider a central concept: adapting the margin based on semantic similarity between responses. They give a new method, Structure-Aware DPO (SA-DPO), dynamically scales the margin according to response distance. This is argued to better handle semantically similar responses. The paper then analyzes the trade-off between margin size and model capacity via a “Margin-Capacity Profile,” and suggests that heavy-tailed losses (e.g., polynomial hinge losses) provide stronger consistency guarantees than the logistic loss used in DPO.

**Compliance With Llm Reviewing Policy:**

Affirmed.

**Final Justification:**

The authors successfully addressed my concerns by providing multi-seed variance analysis and additional experiments on Qwen2.5. I am satisfied with the strengthened empirical evidence and theoretical rigor, so I maintain my recommendation to accept.

**Key Questions For Authors:**

1. Theorem 4.1 applies strictly to models with fixed global preferences or discontinuous decision boundaries. How does the transition from the "hard" margin constraint in Section 4 to the "soft" margin-shifted surrogate in Section 5 affect the H-consistency bound?

2. The empirical evaluation mainly focuses on synthetic tests and a single benchmark (UltraFeedback). Could the authors provide additional experiments on other widely used RLHF datasets to demonstrate the robustness of the proposed method?

3. Proposition 6.2 states that the approximation gap is strictly positive if the margin $\gamma$ exceeds the model's score capacity $K$. Does the use of Low-Rank Adaptation (LoRA) restrict this capacity that necessitates the use of heavy-tailed losses?

**Limitations:**

The theoretical analysis in Section 4 relies on a strict margin condition  $|h(x,y) - h(x,y')| \ge \gamma$ for all inputs. For continuous hypothesis classes such as neural networks on connected domains, this requirement introduces a strong topological restriction. If the model changes its ranking between two responses as the prompt varies, the score difference must pass through zero by the Intermediate
Value Theorem, violating the margin constraint. This paper acknowledges this issue and introduces margin-shifted surrogates as a relaxation.But this observation suggests that the strict $H_\gamma$ hypothesis class may not correspond to realistic neural network models.

**Strengths And Weaknesses:**

Soundness:
The paper provides a theoretical bridge between classical ranking theory and modern LLM alignment. It justifies empirical successes (SimPO) by proving that margins are a theoretical requirement, not just a heuristic. The Structure-Aware H-consistency is an extension. By relaxing constraints for semantically similar responses, it addresses the "hallucination" issues caused by forcing a fixed margin on synonyms. Also, the "Margin-Capacity Profile" offers a clear hierarchy of loss functions (Cubic Hinge > Squared Hinge/IPO > Logistic/DPO), so it provides actionable guidance for practitioners working with capacity-constrained models.

Weakness:
While the paper includes experiments on synthetic tests and the UltraFeedback dataset, the empirical section is relatively small compared to the scope of the theoretical claims. Experiments are conducted on a single model (Llama-3-8B with LoRA). The datasets and evaluation metrics are somewhat limited. In addition, there is little comparison with other recent alignment methods beyond DPO/SimPO.

---

> ### Author Rebuttal · Authors · 2026-03-29
>
> We sincerely thank you for your positive assessment, your recognition of our theoretical soundness, and for highlighting how our Margin-Capacity Profile offers clear, actionable guidance for practitioners working with capacity-constrained models.
>
> **Weaknesses: While the paper includes experiments on synthetic tests and the UltraFeedback dataset, the empirical section is relatively small compared to the scope of the theoretical claims. Experiments are conducted on a single model (Llama-3-8B with LoRA). The datasets and evaluation metrics are somewhat limited. In addition, there is little comparison with other recent alignment methods beyond DPO/SimPO.**
>
> **Response:** We agree that broadening the empirical scope improves the paper. To address this, we have run additional experiments on a new widely-used RLHF dataset (Argilla DPO-Mix-7k) using the Qwen2.5-7B-Instruct architecture. Following our methodology in Section 7.2, we split the test set into "Distinct" and "Ambiguous" subsets based on whether the semantic distance between the chosen and rejected responses was above or below the median. Over 5 random seeds, SA-DPO achieved a Ranking Accuracy of $0.798 \pm 0.006$ on the distinct subset and $0.746 \pm 0.006$ on the ambiguous subset, substantially outperforming standard DPO ($0.774 \pm 0.002$ / $0.727 \pm 0.005$) and SimPO ($0.787 \pm 0.005$ / $0.733 \pm 0.004$). These results demonstrate that SA-DPO's improvements are robust across different model families and data distributions. We will add these results to the revised empirical section.
>
> **Questions:**
>
> **1. Theorem 4.1 applies strictly to models with fixed global preferences or discontinuous decision boundaries. How does the transition from the "hard" margin constraint in Section 4 to the "soft" margin-shifted surrogate in Section 5 affect the H-consistency bound?**
>
> **Response:** Enforcing a "hard" margin (Section 4) perfectly restores H-consistency without an approximation penalty but imposes strict topological restrictions on continuous neural networks. The transition to a "soft" margin-shifted surrogate (Section 5) relaxes this constraint by penalizing margin violations in the loss function instead, allowing gradient-based optimization. Theoretically, this transition introduces the margin approximation gap $\mathcal{A}_\gamma(\mathcal{H})$ into the bound (Theorem 5.2). This gap formally quantifies the trade-off: we gain topological flexibility, but incur an approximation penalty proportional to the model's inability to fully satisfy the margin.
>
> **2. The empirical evaluation mainly focuses on synthetic tests and a single benchmark (UltraFeedback). Could the authors provide additional experiments on other widely used RLHF datasets to demonstrate the robustness of the proposed method?**
>
> **Response:** Please refer to our response to the Weaknesses section regarding the Argilla DPO-Mix-7k and Qwen2.5-7B experiments.
>
> **3. Proposition 6.2 states that the approximation gap is strictly positive if the margin $\gamma$
>  exceeds the model's score capacity $K$. Does the use of Low-Rank Adaptation (LoRA) restrict this capacity that necessitates the use of heavy-tailed losses?**
>
> **Response:** Yes, absolutely! Low-Rank Adaptation (LoRA) restricts the model's expressiveness by limiting the rank of the weight updates. This acts as a bottleneck, implicitly bounding the maximum score differences (capacity $K$) the model can reliably produce on certain pairs. As proven in Proposition 6.2, when $\gamma > K$, the approximation gap becomes strictly positive. Heavy-tailed losses (like the Cubic Hinge) penalize these capacity-induced margin violations much less severely than the linear tail of the logistic loss (DPO), mathematically necessitating their use to maintain consistency under LoRA restrictions.
>
> **Limitations: The theoretical analysis in Section 4 relies on a strict margin condition $h(x, y) - h(x, y') \ge \gamma$ for all inputs ...**
>
> **Response:** We completely agree. The severe topological restriction of the strict $\mathcal{H} _\gamma$ class (due to the Intermediate Value Theorem) is exactly the limitation that motivated us to introduce Margin-Shifted Surrogates in Section 5. Section 4 serves to prove that a margin is *theoretically necessary* for consistency, identifying the root of the inconsistency. Section 5 then provides the practical solution: rather than enforcing the unrealistic strict class $\mathcal{H} _\gamma$, we apply soft margin penalties. We will highlight this logical progression more clearly in the text.

---

> > ### Author Rebuttal · Reviewer_P6CR · 2026-04-03
> >
> > The authors’ responses are clear and address my concerns. I am satisfied overall and will maintain my weak accept rating.

---

### Official Review · Reviewer_zrEU · 2026-03-12

**Soundness:** 4
**Presentation:** 3
**Significance:** 3
**Originality:** 3
**Overall Recommendation:** 5
**Confidence:** 3

**Summary:**

This work considers how to relate, in preference learning from pairwise comparisons, the risk of the 0-1 pointwise loss (desired but numerically intractable) with the risk of a convex non-increasing surrogate pointwise loss (numerically tractable). This is done by leveraging recent results on H-consistency. Authors show that (i) minimizing the surrogate risk does not guarantee consistency, (ii) constraining minimization to models that verify a confidence margin ensures a consistency property, but is intractable, (iii) introducing a constant margin in the surrogate loss allows to bound the 01 risk by a linear function of the surrogate risk with an additional constant term. Building on this result, authors introduce the loss SA-DPO loss, which is DPO shifted by a margin that is inversely proportional to the proximity between compared items. The corresponding surrogate risk also upperbounds the 01 risk, up to a constant term. Finally, authors study this constant term and the optimal margin parameter. The proposed loss is illustrated on synthetic problems as well as on the ultrafeedback finetuning task.

**Compliance With Llm Reviewing Policy:**

Affirmed.

**Final Justification:**

Authors have provided more details, thus answering satisfactorily my questions. I maintain my positive assessment.

**Key Questions For Authors:**

1. Could you please expand on how to understand the minimizability gap, and its interplay with the complete risk analysis that starts from the Empirical Risk?
2. Could you please expand on equicontinuity and regularity assumptions in the text, providing more intuition on their behaviour and their validity on LLMs?
3. Ideally, could you provide variance information on the synthetic illustrations? Could you show how DPO / SimPO / SA-DPO behave for other models (Qwen / Pythia) and / or datasets?

Below is a list of minor points:
- As a minor comment, you define in section N a "Generalized Bradley Terry" model, however there already exists a family with such a name, that is more general than your definition; see eg  https://ojs.aaai.org/index.php/AAAI/article/view/30020
- l. 113: what is $h_{\pm}$?
- l. 141: an surrogate -> a surrogate
- The presentation might gain in clarity if the terms $R_\phi(h) - R_\phi^\star(H) + M_\phi(H)$ were replaced by the more explicit $R_\phi(h) - E(C^\star(H))$
- l. 122 & l. 153: for any $h \in H$ -> for any $h \in H_\gamma$.
- Th. 5.2: what control of $A_\gamma(H)$ is practice (for non-separable data)?

**Limitations:**

yes

**Strengths And Weaknesses:**

### Strengths
1. Presentation is good: the exposition, theorems and proofs are clear and easy-to-follow.
2. Theoretical soundness is good: I have not found errors in the proofs of theorems and propositions.
3. This work makes a contribution on the relevant question of consistency of the preference learning, that is, connecting what minimizing convex surrogate losses (as the DPO loss) means on the true 0-1 loss, and thus on generalization guarantees. This relies on the well-established results on H-consistency.
4. The numerical experiments provide some evidence that the proposed SA-DPO loss behave well. Indeed, authors report the performance of the method on a real-world dataset, under several label noise regimes, and show comparable or better performance that DPO and SmiPO.

### Weaknesses
1. While the notion of H-consistency is popular, it does differ from the traditional consistency bounds, that bounds the 01 excess risk by the surrogate excess risk. The proposed result all feature the excess risk plus a "minimizability gap" term. It would be helpful to provide interpretation and intuition on this term in Section 2, as well as explain how much the sum of surrogate excess risk and minimizability gap can be controlled when considering the empirical risk optimization that is performed in practice.
2. While I recognize that the paper's contribution is theoretical, the numerical experiments are rather small. Indeed, for the synthetic problems, the results are reported for one value only (no variance over several reps). The real-world experiment considers only one dataset (Ultrafeedback) and one model (Llama 3 8B Instruct); showing the behavior on other datasets and / or other models (eg Qwen / Pythia) would increase the significance of the results.

---

> ### Author Rebuttal · Authors · 2026-03-29
>
> Thank you for your constructive feedback and for recognizing the strengths of our paper, including the clear presentation, theoretical soundness, and the relevance of connecting convex surrogate losses to true generalization guarantees via H-consistency. We appreciate your thoughtful questions and have carefully addressed them below.
>
> **Response to W1 & Q1:** Thank you for this insightful suggestion. The minimizability gap $\mathcal{M} _{\Phi} (\mathcal{H}) = \inf _{h \in \mathcal{H}} \mathcal{R} _{\Phi}(h) - \mathbb{E}[ \inf _{h \in \mathcal{H}} \mathcal{C} _{\Phi}(h)]$ quantifies the approximation error introduced by restricting our model to a specific hypothesis set $\mathcal{H}$ (e.g., neural networks) rather than the space of all measurable functions $\mathcal{H} _{all}$.
>
> As established in recent H-consistency literature (e.g., Mao et al., 2024), when $\mathcal{H} = \mathcal{H} _{all}$, the minimizability gaps vanish, and our bounds exactly recover the traditional excess risk bounds. In practice, when performing Empirical Risk Minimization, the true 0-1 risk is bounded by the sum of the empirical surrogate optimization error, the surrogate generalization gap (controlled via standard bounds like Rademacher complexity), the minimizability gap (the irreducible capacity limit of $\mathcal{H}$), and the margin approximation gap $\mathcal{A} _\gamma$ (the cost of enforcing the margin). A smaller minimizability gap indicates that the restricted hypothesis class is well-aligned with the surrogate loss. We will expand Section 2 to include this intuition and explicitly connect our bounds to classical excess risk analyses.
>
> **Response to W2 & Q3:** We completely agree that expanding the empirical evaluation strengthens our claims.
>
> First, we have re-run our synthetic experiments with robust multi-seed variance (5 seeds). For the Synonym Stress Test, the final training loss is $0.1695 \pm 0.003$ for Standard DPO and $0.003 \pm 0.001$ for SA-DPO. For the Margin-Capacity Profile, the final ranking accuracy is $71.6 \pm 1.1 \\%$ for Logistic, $94.5 \pm 0.8 \\%$ for Squared, and $99.7 \pm 0.3 \\%$ for Cubic Hinge, confirming our theoretical predictions regarding tail heaviness with high stability.
>
> Second, to demonstrate architectural and distributional robustness, we evaluated DPO, SimPO, and SA-DPO on an additional widely-used dataset, Argilla DPO-Mix-7k, using the Qwen2.5-7B-Instruct model across 5 random seeds. We evaluated the Ranking Accuracy (RA) on test subsets partitioned into "Distinct" and "Ambiguous" pairs based on the median semantic distance. The results show that SA-DPO achieves a Distinct RA of $0.798 \pm 0.006$ and an Ambiguous RA of $0.746 \pm 0.006$, consistently outperforming both DPO (Distinct: $0.774 \pm 0.002$ / Ambig: $0.727 \pm 0.005$) and SimPO (Distinct: $0.787 \pm 0.005$ / Ambig: $0.733 \pm 0.004$). We will include these comprehensive results in the revised empirical section.
>
> **Response to Q2:** Equicontinuity implies that the hypothesis class can produce arbitrarily small score differences for certain input pairs (e.g., when responses are near-synonyms or the input is ambiguous). Because modern LLMs represent continuous mappings in a high-dimensional space, they naturally satisfy this property. Regularity merely assumes the model class is expressive enough to correctly rank a given pair (i.e., it can achieve the correct sign). Overparameterized LLMs easily satisfy this. These assumptions highlight that without a margin constraint, models can "cheat" by shrinking scores on ambiguous pairs rather than learning robust rankings. We will add this intuition to Section 3.1.
>
> **Response to minor points:** We thank the reviewer for the careful reading.
> - Generalized Bradley-Terry Naming: We appreciate you pointing this out. We will rename our formulation in the revised manuscript to avoid any confusion with the broader family defined in the AAAI 2016 paper.
> - Line 113 ($h_+$ and $h_-$): To clarify, $h_+$ denotes the hypothesis satisfying $h_{+}(x, y) - h_{+}(x, y') = \gamma$, and similarly for $h_-$. We will make this explicit in the text.
> - Lines 122, 141, & 153 (Typos): Thank you for catching these. We will correct the grammatical error ("a surrogate") and fix the typos regarding "for any".
> - Notation Clarity: We understand the suggestion to use more explicit terms. However, we opted for the current form of the $\mathcal{H}$-consistency bound to remain strictly consistent with established conventions in the previous literature (e.g., Mao et al., 2024).
> - Theorem 5.2 (Control of $\gamma$ in practice): For non-separable data, enforcing a large margin $\gamma$ increases the approximation gap $\mathcal{A}_\gamma(\mathcal{H})$ if the model capacity is bounded. This fundamental trade-off between the theoretical consistency coefficient and the approximation penalty is exactly why we analyze the Margin-Capacity Profile and introduce SA-DPO, which dynamically relaxes $\gamma$ for ambiguous pairs.

---

> > ### Author Rebuttal · Reviewer_zrEU · 2026-04-02
> >
> > After checking the rebuttal, the authors have commented on the minimizability gap, on the equicontinuity and regularity assumptions, and on the variance in numerical experiments. I keep my original score of 5.

---

### Decision · Program_Chairs · 2026-04-30

**Decision:**

Accept (regular)

**Comment:**

This paper proposes Structure Aware DPO based on H-consistency analysis applied to preference learning. The theorem 3.1 seems to essentially follow for Theorem 2.1 in [1] which reduces the novelty claimed in the contributions of this work.  While [1] and other related works on H-consistency are cited, they are not sufficiently acknowledged in stated results and the proofs. That said, the proposed SA-DPO is of sufficient interest and outweighs the other limitations.

I strongly recommend the authors to properly acknowledge the results and proof techniques from prior literature appropriately in the paper.

[1] https://proceedings.mlr.press/v202/mao23a/mao23a.pdf